# Graph State Networks (GSNs):
# Persistent Nodewise Selective State Space Models

**Arijit Dey**                                                                                    *arijit.dey.1729@gmail.com*

**Bahman Gharesifard**                                                                            *bahman.gharesifard@queensu.ca*
*Queen's University, Canada*

**Reviewed on OpenReview:** *https://openreview.net/forum?id=zMEuBQfeT6*

## Abstract

Temporal graphs are often observed as streams of timestamped interactions, where accurate prediction requires retaining and selectively using historical information nodes. Existing temporal graph models either (i) recompute representations from a sliding neighborhood/history at query time, or (ii) maintain a memory module but offer limited control and limited theory for what is retained over long horizons. We propose **Graph State Networks (GSNs)**, a bucketed temporal-graph framework that maintains a persistent hidden state per node and updates it online using a content- and time-dependent selective state space update. Concretely, GSNs store node states in an explicit id-indexed state table and for each bucket, *read* the current state, *update* it with a time-aware Mamba-like mechanism, and *commit* the state back via an exponential moving average controlled by a commit-rate. This commit mechanism provides an explicit "retention dial" and enables a tractable analysis of forgetting. We develop a *capacity/recall theory* for persistent node memory and show that, under *incremental-stability* assumptions on blank-bucket dynamics, the influence of a single past event admits a geometric forgetting bound, with the effective decay governed by the contraction of the blank dynamics and the commit mechanism. Empirically, GSNs are competitive on standard dynamic link prediction benchmarks under *Average Precision (AP)*, with the strongest gains appearing in several inductive settings, while AUC-ROC remains more mixed. We validate these ideas with controlled synthetic write–wait–read probes. Under shared later blank sequences and a small nonzero state-noise blank update, the measured write-vs-zero-write relative influence exhibits near-exponential decay over the main operating regime. Our simulation studies verify this overall trend, including the nonzero floor at larger commit rates. We provide an extended implementation of GSNs[1].

## 1 Introduction

Many real-world relational systems, e.g., communication networks, collaboration graphs, recommender systems, transaction ledgers, are naturally modeled as *temporal graphs*, where edges arrive as timestamped events. A core problem is **dynamic link prediction**: given past interactions, score which edges are likely to occur next. The challenge is not only modeling graph structure, but doing so *online* while handling long and irregular histories.

A common design choice is *where the history is stored*. In one family of methods, node representations are computed (or recomputed) from a window of recent interactions using message passing and/or attention over sampled temporal neighborhoods. This can be effective, but it inherits familiar limitations of deep message passing–difficulty propagating long-range signals due to bottlenecks ("over-squashing") and representation

---

[1]https://github.com/arijitcodespace/GSN

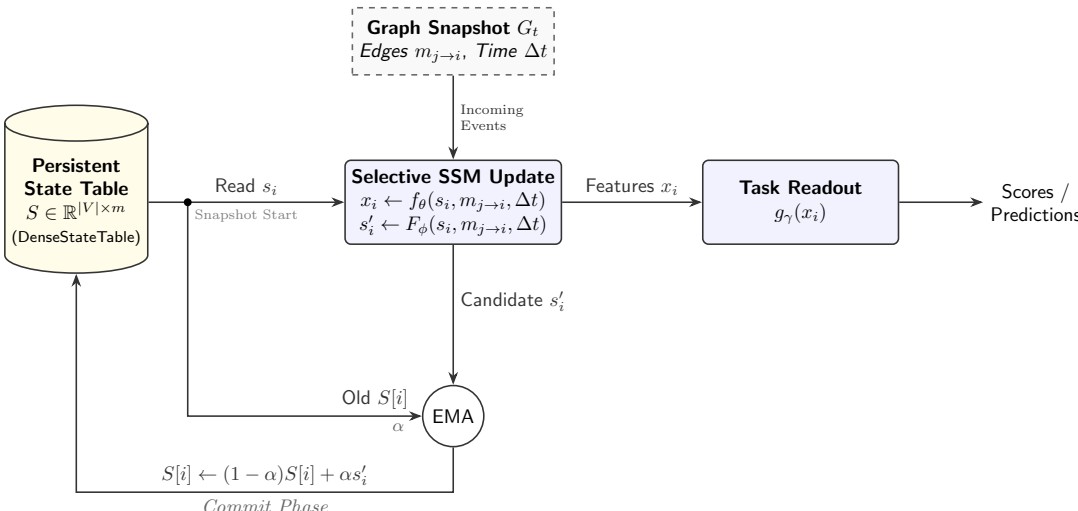

Figure 1: GSN Layer (*read–update–commit*) for a temporal graph snapshot. Here $V$ is the set of nodes and $m$ is the state dimension. Each node $i$ has a persistent state $s_i \in \mathbb{R}^m$ stored in an ID-indexed persistent state table $S \in \mathbb{R}^{|V| \times m}$ that carries across buckets (hence persistent). At snapshot start, the layer reads $s_i$, then uses incoming messages/events $m_{j \to i}$ and elapsed time $\Delta t$ in a time-aware selective state-space update to produce features $x_i$ for the task head $g_\gamma(x_i)$ and a proposed new state $s_i'$. After scoring, the new states are committed to the table via an Exponential Moving Average (EMA) with commit rate $\alpha$ using (1).

collapse in deep stacks ("over-smoothing"), while neighborhood retrieval can become expensive or brittle under distribution shift. A second family uses more global attention-style mechanisms (temporal Transformers and graph Transformers), which can capture long-range dependencies but often lead to quadratic costs in sequence length or require substantial positional/structural encoding machinery.

A different viewpoint is to treat each node as an **online state machine**. In particular, instead of rebuilding a node embedding from scratch at query time, a persistent state is updated as events arrive. This viewpoint is present in several temporal-graph architectures (e.g., memory module in Temporal Graph Networks), but two gaps remain in practice:

1. **Update mechanism:** Many memory updates are hand-designed (GRU-style) or tied closely to specific neighborhood aggregation schemes. Recent progress in **selective state-space models (SSMs)**, notably Mamba (Gu & Dao, 2023), suggests that content-dependent, gated state updates can provide strong long-range modeling with favorable scaling.

2. **Controllability and interpretability of forgetting:** Even when a model maintains memory, it is often unclear *what* information persists and how *quickly* it decays as time passes without a new signal. This plays a key role in deployment, because while some applications need long retention, others need rapid adaptation.

This work proposes **Graph State Networks (GSNs)** as a step toward addressing both gaps. GSNs make a simple but strict architectural commitment: every node $i$ owns a persistent hidden state $s_i \in \mathbb{R}^m$ that is **not reinitialized per batch or per forward call**, but evolves over the temporal stream. Concretely, we maintain an explicit state table $S$ indexed by global node IDs. For each temporal bucket/snapshot, we:

- **Read** the current persistent state for nodes present in the bucket;

- **Update** it using a time-aware Mamba-like block;

- **Write/commit** the new state back to the table using an Exponential Moving Average (EMA) with commit rate $\alpha \in (0, 1]$:

$$S[\text{ids}(i)] \leftarrow (1 - \alpha)S[\text{ids}(i)] + \alpha s_i, \tag{1}$$

where $s_i$ is the calculated *new state* of the node $i$, and $\text{ids}(i)$ is the ID of node $i$ as a tuple of $(a, b)$ indexing the row and column of $S$.

This design cleanly separates (i) *how* information is integrated into a node's memory (the selective SSM update) from (ii) *how aggressively* that memory is refreshed (the commit rule). It also matches our implementation setting: we use bucketed processing, where node states persist across buckets and are advanced via explicit commit steps at bucket boundaries, rather than strict event-by-event streaming updates. This introduces a practical tradeoff: smaller buckets more closely approximate streaming and preserve finer temporal order, but require more frequent state updates and substantially higher training/inference cost; larger buckets reduce the number of commits and forward passes, but coarsen temporal resolution. The strict streaming regime is recovered as the special case when bucket size equals one.

Beyond architecture, our second goal is to make node retention mechanism measurable. We develop a capacity/recall theory for persistent node states.In particular, since a fixed-dimensional state must compress history, it cannot reproduce arbitrary query-time retrieval, but its retention can be characterized in terms of what remains recoverable after many events. A key object is the model's behavior during **blank buckets**, i.e., periods with no new informative input, where the system's intrinsic dynamics determine whether old information decays, persists, or is overwritten by drift. Under incremental stability assumptions for time-varying blank-bucket dynamics, we obtain geometric forgetting bounds for state differences. We emphasize that this theory is intended as a lens on retention in persistent node states rather than as a complete characterization of the full trained model on benchmark tasks. In particular, the blank-bucket analysis isolates a regime in which no new task-relevant signal arrives, allowing the contribution of the commit rule and the intrinsic state dynamics to be studied separately from the full predictive setting. Our empirical probes are designed to test whether this lens is informative for the learned model. For the learned blank operator used in experiments, we additionally fit a simple empirical stepwise recursion to summarize observed influence decay.

**Contributions.** In summary, this work:

- introduces **Graph State Networks**, a persistent node-state architecture updated by a time-aware selective SSM and committed via an EMA rule with rate $\alpha$;

- develops a **blank-bucket forgetting analysis** that yields theory-guided bounds on single-event influence decay under incremental-stability assumptions;

- proposes and evaluates **synthetic write–wait–read probes** that diagnose what the node state retains and show that, in the studied synthetic setting, short-horizon measurements can be used to predict longer-horizon retention trends;

- shows that a persistent node-state architecture can remain competitive on standard temporal link prediction benchmarks under AP, and at the same time exposes an explicit retention dial through the commit rate $\alpha$.

## 2 Related Work

A large body of literature studies representation learning on temporal interaction graphs, e.g., (Rossi et al., 2020; Wang et al., 2021b; Xu et al., 2020). Early and influential approaches include **JODIE** (Kumar et al., 2019), which learns dynamic embedding trajectories for interacting entities, and **DyRep** (Trivedi et al., 2019), which models evolving node representations driven by communication/association processes in continuous time. Attention-based temporal neighborhood aggregation was popularized by **TGAT** (Xu et al., 2020), which introduces functional time encodings and temporal self-attention for inductive temporal

graphs. **Temporal Graph Networks (TGN)** (Rossi et al., 2020) unify several lines of work by combining a memory module with message-passing operators over temporal neighborhoods.

Beyond neighbor aggregation, motif/walk-based and Transformer-style models have been explored. **CAWN** (Wang et al., 2021a) uses causal anonymous walks to capture temporal motifs in an inductive way. **TCL** (Wang et al., 2021b) adapts Transformer-style modeling to temporal graphs with contrastive learning objectives. More recently, a line of work argues that carefully designed but simpler architectures can be highly competitive: **GraphMixer** (Cong et al., 2023) proposes an MLP-centric design for temporal link prediction, while **DyGFormer** (Yu et al., 2023) uses a Transformer with co-occurrence encoding and patching to leverage longer histories efficiently. Benchmarking efforts such as the **Temporal Graph Benchmark (TGB)** emphasize realistic evaluation protocols and include strong non-parametric baselines like **EdgeBank** (Huang et al., 2023).

**State-space models for sequences and graphs:** Structured state space models such as **S4** (Gu et al., 2021) demonstrated that long-range sequence dependencies can be modeled efficiently with principled linear dynamical systems. **Mamba** (Gu & Dao, 2023) introduced *selective* (input-dependent) state updates that achieve strong empirical performance with linear-time scaling in sequence length. Subsequent work like **Mamba-2** (Dao & Gu, 2024) further clarified connections between attention and structured state space computation.

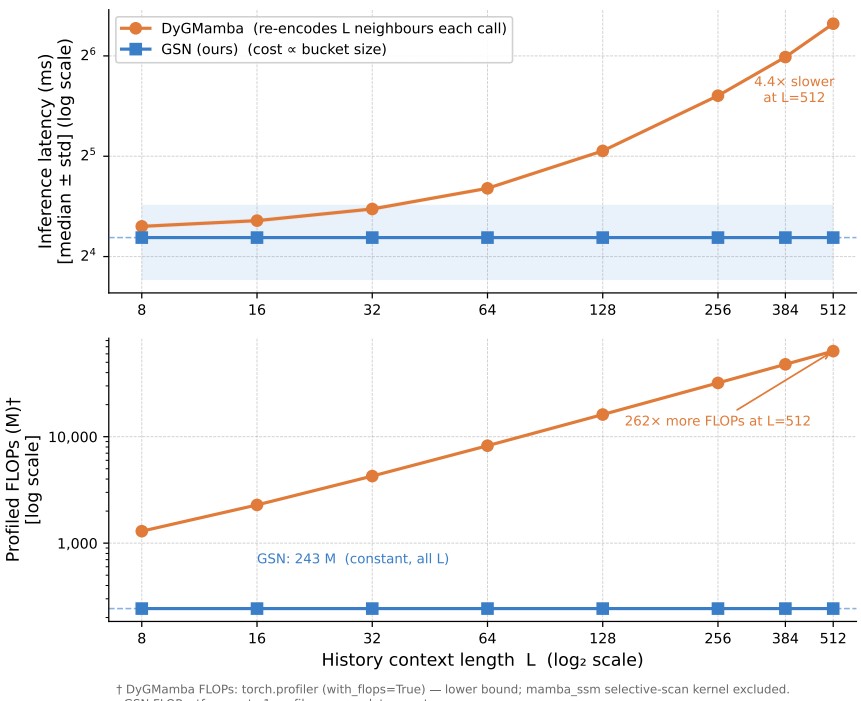

Figure 2: **Inference cost vs. history context length** $L$. *Top*: median latency $\pm$ std over 50 runs. DyGMamba (CUDA) latency grows $O(L)$ + neighbor lookup as it re-encodes the last $L$ temporal neighbors per call; (the constant overhead (neighbour-sampler lookup) dominates at small $L$, causing the slight sub-linear appearance at low $L$ on the log–log scale; at large $L$ the $O(L)$ compute term dominates) an $O(L)$ linear fit (dashed) tracks the trend closely. GSN (CUDA) latency is constant. *Bottom*: profiled FLOPs (log scale). At $L = 512$, DyGMamba requires $262\times$ more FLOPs than GSN. GSN FLOPs are constant at 243 M for all $L$, measured via `tf.compat.v1.profiler` (complete count). DyGMamba FLOPs are a lower bound (`torch.profiler`; `mamba_ssm` selective-scan kernel excluded). Batch size 200; hidden dimension 256 for both models; Wikipedia node/edge feature dimensions.

Motivated by these, several papers adapt SSM ideas to graph-structured learning. **S4G** (Song et al., 2024) introduces structured state spaces into graph settings to address long-range dependencies while retaining useful inductive bias. **DyGMamba** (Ding et al., 2025) adapts Mamba-style selective state space models to **continuous-time dynamic graphs** by encoding per-node interaction histories with an SSM and using an additional time-level SSM to select critical historical information, achieving strong dynamic link prediction performance with favorable efficiency. **Graph Mamba Networks** (Wang et al., 2024) propose a general framework that sequentializes graph neighborhoods and processes them with bidirectional selective SSM encoders. These approaches often focus on static graphs or on building graph-to-sequence encoders within each forward pass.

**Positioning.** GSNs introduced in this paper target the same temporal link prediction setting and evaluation protocol used by these baselines. Conceptually, GSNs are related to memory-based temporal GNNs (e.g., TGN) in that they maintain persistent node memories, but they differ in two important ways: (i) the memory update is implemented as a selective state-space update (SSM) rather than a purely GRU/message-passing update, and (ii) we explicitly analyze forgetting/retention induced by the commit mechanism and blank dynamics, rather than treating the memory as a black box.

The intended contribution is therefore not merely to replace a GRU-style updater with an SSM-style updater. Rather, GSN is built around an explicit read–update–commit semantics in which the persistent state is the primary object of interest; the updater ablation in Table 5 provides empirical support that the selective SSM component also matters in practice.

As far as SSM-like models are concerned, GSNs use selective SSM machinery but elevate the "sequence position" notion to **node identity**: the state is persistent per node and stored in an explicit table. Unlike many graph-SSM encoders that reconstruct representations from a context window, GSNs are designed around **state persistence across temporal buckets** with explicit read/update/commit semantics. This allows us to ask and *partially* study a different question: *"can SSMs work on graphs?"* and *"what does a persistent node state retain, and how can we measure and control its forgetting"*?

## 3 Preliminaries: Persistent Node State Dynamics in GSNs

This section formalizes the state evolution semantics that the rest of the theory studies. The key design choice is that **each node owns a persistent hidden state** that is **stored outside** the forward pass and **updated/committed** only at well-defined times in a bucketed protocol.

### 3.1 Temporal interaction stream and bucketed protocol

We observe a temporal interaction stream of events $\{e_k\}_{k=1}^N$, where each event $e_k = (u_k, v_k, t_k, x_k)$ contains an interacting pair $(u_k, v_k)$, a timestamp $t_k$, and optional *edge* features $x_k$. Let $V$ be the global node set and ids($i$) map to a stable integer ID.

We partition time into $K$ chronological buckets $B^{(1)}, \ldots, B^{(K)}$. For bucket $B^{(k)} = [t_{\text{start}}^{(k)}, t_{\text{end}}^{(k)}]$, we distinguish:

- **History snapshot (pre-bucket):** The graph $G_{\text{prev}}^{(k)}$, containing all events with timestamp $< t_{\text{start}}^{(k)}$.

- **Commit snapshot (up to bucket end):** The graph $G_{\text{end}}^{(k)}$, containing all events with timestamp $\leq t_{\text{end}}^{(k)}$.

This separation is important for leakage-free evaluation: we want predictions inside bucket $k$ to depend only on history before the bucket, while still allowing the model's persistent memory to be updated with the bucket's events after scoring. The bucket size is therefore an explicit operating-point choice. Finer buckets preserve more temporal detail and bring the protocol closer to true streaming updates, but they also increase the number of read-update-commit cycles executed over the event stream. Coarser buckets amortize computation across more events and reduce update frequency, at the cost of lower temporal resolution within each bucket.

### 3.2 Persistent state table and read–update–commit

A GSN maintains a **persistent state table** $S \in \mathbb{R}^{|V| \times m}$, where the row $S[\mathrm{ids}(i)]$ is node $i$'s memory/state (dimension $m$). This table persists across buckets and is conceptually a part of the model's state.

For a given snapshot (e.g., $G_{\mathrm{prev}}^{(k)}$ or $G_{\mathrm{end}}^{(k)}$), the model performs:

1. **Read:** for each node $i$ participating in the snapshot, read $s_i \leftarrow S[\mathrm{ids}(i)]$.

2. **Pre-commit update:** produce a proposed new state $s_i'$ via a parameterized update operator $F_\phi :$ $\mathbb{R}^m \times \mathcal{E} \times [0, T] \mapsto \mathbb{R}^m$ such that

$$s_i' = F_\phi(s_i; \text{snapshot context}, \Delta t).$$

Here, $m$ is the state dimension, $\mathcal{E}$ is the edge set (snapshot context denotes the context encoded within edges in a given snapshot), $\Delta t \in [0, T]$ denotes elapsed time since node $i$ was last updated, and $T$ denotes the last timestamp of the data (See Figure 1). In our implementation, this update is realized with a time-aware selective state-space block (Mamba-style), but the theory below only requires that it induces some map $F_\phi$ on states within a bucket.

**Commit (persistent write):** update the table using an exponential moving average with commit rate $\alpha$:

$$S[\mathrm{ids}(i)] \leftarrow (1 - \alpha)S[\mathrm{ids}(i)] + \alpha s_i',$$

where $\alpha \in (0, 1]$ can be fixed or scheduled.

Intuitively, the pre-commit update computes "what the node state should become given the bucket", while the commit step controls how aggressively that proposal overwrites persistent memory. The scalar $\alpha$ will reappear in the theory as a direct knob on forgetting and mixing.

### 3.3 Stateful training/evaluation semantics with no leakage

Within bucket $B^{(k)}$, we use two-snapshot protocol:

- **Score step:** run the model on $G_{\mathrm{prev}}^{(k)}$ to obtain node representations, then score candidate edges whose timestamp lies in bucket $B^{(k)}$.

- **Commit step:** after scoring, update/commit the persistent states using $G_{\mathrm{end}}^{(k)}$, so that the next bucket starts from memory that has incorporated bucket-$k$ interactions.

**Clarification of the two pass bucket update:** In our implementation, both passes mutate the persistent state table $S$. First, training runs on the pre-bucket history snapshot with commit enabled; this is a real write to $S$. Because $G_{\mathrm{prev}}^{(k)}$ contains only events with timestamps $< t_{\mathrm{start}}^{(k)}$, the first write cannot leak bucket-$k$ information into the scores. Its role is to align the persistent table with the history used for scoring. Second, after the bucket-$k$ candidates have been scored, we do a *second commit* which performs a *second* real write using the snapshots up to $t_{\mathrm{end}}^{(k)}$, so that the state table carried into bucket $k + 1$ reflects all interactions up to the end of bucket $k$. Thus there are **two real commits per bucket, on two different snapshots, with two different purposes:** leakage-free scoring and state carry-forward.

This makes the stateful model well-defined: predictions for bucket $k$ are conditioned on history strictly before the bucket, but the state entering the bucket $k + 1$ reflects everything up to $t_{\mathrm{end}}^{(k)}$.

### 3.4 Quantities the theory will track

In this section, we aim to study: **what does a bounded-dimensional persistent state retain, and how does it forget?** To make this precise, we define three objects:

1. **The committed bucket operator.** Fix a bucket snapshot structure and the exogenous inputs it induces, e.g., features, time deltas, etc. The combination "pre-commit update + EMA commit" defines an operator $T_\alpha$ on node states: $s \mapsto (1-\alpha)s + \alpha F_t(s)$, where $F_t$ is a time-varying pre-commit update map induced by that bucket. In the next section, we study this operator in the special but revealing case of "blank buckets".

2. **Influence of a write (counterfactual sensitivity).** Consider two executions that are identical in all later blank buckets but differ in the write initialization at time 0 (for example, an actual write versus a zero-write baseline, or two different write payloads). Let $s_t$ and $\hat{s}_t$ be the resulting node states after $t$ subsequent bucket commits under identical later inputs. **The influence at delay** $t$ is measured by $\|s_t - \hat{s}_t\|$. This is the central object for "forgetting".

3. **Recall through a readout.** A downstream task uses a decoder/readout $g$ (e.g., link predictor or probe head) applied to the node state. We will bound not just state differences, but also readout differences $\|g(s_t) - g(\hat{s}_t)\|$ under mild regularity assumptions, connecting "state retention" to "task-visible retention".

With these definitions in place, the next section derives explicit convergence and geometric forgetting rates for repeated blank-bucket commits, and shows how $\alpha$ and the blank dynamics jointly determine memory half-life.

## 4 Blank-Bucket Theory: Forgetting and State Retention Bounds

This section formalizes the "retention dial" perspective of the commit rate $\alpha$ by analyzing what happens when a node experiences *no new informative input* for multiple buckets.

### 4.1 Blank buckets

A **blank bucket** is a bucket in which the node receives no new task-relevant signal, e.g., features are zeroed, or messages carry a fixed "blank token", but the model is still executed and the resulting state is committed to the persistent state table (as in the write–wait–read protocol).

Let $s_t$ be the persistent state of a fixed node after $t$ blank-bucket commits following some "write" event. The GSN update semantics from the previous section imply a committed recursion of the form:

- pre-commit proposal: $s \mapsto F_t(s)$ (this is "whatever the backbone does on a blank bucket")

- commit: $s_{t+1} = (1-\alpha)s_t + \alpha F_t(s_t)$

### 4.2 Main result: closed form influence decay

**Setup and Assumptions**  Let $m, n \in \mathbb{Z}_{>0}$. For all $t \geq 0$, let $D_t \subseteq \mathbb{R}^m$ be a convex set. Consider a time-varying discrete time system with an exogenous input/disturbance

$$f : \mathbb{R}^m \times \mathbb{R}^n \to \mathbb{R}^m, \qquad x_{t+1} = f_t(x_t, u_t), \qquad t \in \mathbb{Z}_{\geq 0},$$

where the states $x_t, y_t \in D_t$ and inputs $u_t, v_t \in \mathbb{R}^n$. Also assume,

$$x_t \in D_t \implies x_{t+1} \in D_{t+1} \quad \text{(forward invariance)}.$$

We assume a time-varying incremental ISS-Lyapunov family $W_t$ in the spirit of the discrete-time $\delta$-ISS Lyapunov inequalities in Bayer et al. (2013, Definition 3), and the time-varying discrete-time ISS Lyapunov families in Laila & Astolfi (2004, Definition 2.3). and say that the system has the incremental ISS-Lyapunov property if there exist constants

$$0 < c_1 \leq c_2 < \infty, \qquad \lambda \in (0, 1),$$

such that

$$\lambda c_2 < c_1,$$

and a class-$\mathcal{K}$ function, (see Sontag (1995, Section 2.1) for a definition), $\sigma : \mathbb{R}_{\geq 0} \to \mathbb{R}_{\geq 0}$ such that for every $t$ and for all $x, y \in D_t$ and all inputs $u, v$ there exists a function $W_t : D_t \times D_t \to \mathbb{R}_{\geq 0}$ such that

$$c_1 \|x - y\|^2 \leq W_t(x, y) \leq c_2 \|x - y\|^2, \tag{2}$$

and

$$W_{t+1}(f_t(x, u), f_t(y, v)) \leq \lambda W_t(x, y) + \sigma(\|u - v\|). \tag{3}$$

A common practical specialization is: $\|u - v\|_\infty \leq \bar{u} \implies \sigma(\|u - v\|_\infty) \leq \bar{d}$, i.e., a bounded additive term.

The next result is a standard consequence of the incremental ISS Lyapunov inequalities above, and provides us with means to characterize the "forgetting" properties of GSNs.

*Theorem* 1. **(Incremental ISS implies exponential forgetting with a floor):** Let $x_{t+1} = f_t(x_t, u_t)$ and $y_{t+1} = f_t(y_t, v_t)$ be two trajectories satisfying *forward invariance* and $u_t, v_t$ be as described above. Suppose (2) and (3) hold. Define

$$\Delta_t = \|x_t - y_t\|_2^2.$$

Then for all $t \geq 0$,

$$\Delta_t \leq \frac{c_2}{c_1} \lambda^t \Delta_0 + \frac{1}{c_1} \sum_{k=0}^{t-1} \lambda^{t-1-k} \sigma(\|u_k - v_k\|), \tag{4}$$

where $\|\cdot\|$ denotes any fixed norm on $\mathbb{R}^n$. In particular, if $\sigma(\|u - v\|_\infty) \leq \bar{d}$ for all $k$, then

$$\Delta_t \leq \frac{c_2}{c_1} \lambda^t \Delta_0 + \frac{\bar{d}}{c_1} \frac{1 - \lambda^t}{1 - \lambda}. \tag{5}$$

Hence,

$$\limsup_{t \to \infty} \Delta_t \leq \frac{\bar{d}}{c_1(1 - \lambda)}.$$

For a proof, see Appendix.

*Remark* 4.1. **(Interpretation):** Theorem 1 does not assert that every trained GSN will exactly forget at a single exponential rate. Rather, it states that *if* the blank-bucket dynamics are incrementally stable, *then* the effect of a past write cannot persist arbitrarily strongly – it contracts geometrically up to a disturbance-dependent floor. In the context of GSNs, this identifies two distinct sources of retention behavior: the contraction induced by the blank dynamics, and the floor induced by persistent perturbations or mismatch between counterfactual trajectories.

Below we show that $\Delta_{t+1}$ can be upper-bounded by $\Delta_t$ as an affine recursion. This will help us choose a suitable recursive model assumption on the *influence* which we define later in the experiments.

With a slight abuse of notation, let $V_t := W_t(x_t, y_t)$. From (3) we have,

$$V_{t+1} \leq \lambda V_t + \sigma(\|u_t - v_t\|).$$

Using (2),

$$V_{t+1} \geq c_1 \|x_{t+1} - y_{t+1}\|^2 = c_1 \Delta_{t+1} \quad \text{and} \quad V_t \leq c_2 \|x_t - y_t\|^2 = c_2 \Delta_t. \tag{6}$$

By substituting (6) into (3), we have that

$$c_1 \Delta_{t+1} \leq \lambda(c_2 \Delta_t) + \sigma(\|u_t - v_t\|)$$
$$\Rightarrow \Delta_{t+1} \leq \underbrace{\frac{\lambda c_2}{c_1}}_{a} \Delta_t + \underbrace{\frac{1}{c_1} \sigma(\|u_t - v_t\|)}_{b_t} = a\Delta_t + b_t. \tag{7}$$

Since, $\lambda c_2 < c_1$, each step in (7) is a contraction.

Next, we discuss how much information GSNs can faithfully recover after $d$ blank commits; we provide a channel-capacity style theorem in what follows.

### 4.3 State-Compression and Recoverable Information

**Setup and Assumptions** Fix a state dimension $m \in \mathbb{N}$. A single node's persistent state at (discrete) time $t$ is a vector $s_t \in \mathbb{R}^m$. A discrete payload/message

$$H \in \{1, 2, \ldots, M\}$$

is written into the state at time $t = 0$ by some write mechanism producing a random initial state $s_0$ whose randomness comes from the randomness of $H$ plus any write randomness.

During the blank bucket period of length $d$, i.e., $d$ blank buckets/commits, the state evolves according to the nonlinear time-varying recursion:

$$s_{t+1} = f_t(s_t), \quad t = 0, 1, \ldots, d-1, \tag{8}$$

where $s_0$ is drawn at random, and each $f_t : \mathbb{R}^m \to \mathbb{R}^m$ is a measurable map[2]. At read time we observe a noisy version of the final state:

$$Y = s_d + Z, \qquad Z \sim \mathcal{N}(0, \sigma^2 I_m), \tag{9}$$

with $Z$ independent of $(H, s_0, \ldots, s_d)$. A decoder $\hat{H} = \mathrm{Dec}(Y)$ attempts to recover $H$.

*Remark* 4.2. **(Justification of noise):** The additive noise model is one standard way to make "how many bits can be stored in $\mathbb{R}^m$" well-posed; without some finite-precision/noise assumption, a real vector can encode arbitrarily many bits.

We make the following assumptions

(C1) **(Message distribution)** $H$ is uniform on $\{1, \ldots, M\}$. The write mechanism induces a conditional distribution $s_0 \mid H$. We absorb any additional write randomness into this conditional law.

(C2) **(Write energy / dynamic range)** The message-dependent variation in the written state is bounded as
$$\mathrm{tr}(\mathrm{Cov}(s_0)) \leq mP \qquad \text{for some } P \geq 0 \tag{10}$$

(C3) **(Uniform contraction of blank dynamics)** Additionally, we assume a uniform contraction as follows and call the effective one-step contraction factor to be $r_t \in (0, 1)$ so that

$$\|f_t(x) - f_t(y)\|_2 \leq r_t \|x - y\|_2 \qquad \forall x, y \in \mathbb{R}^m, \quad \forall t \in \{0, \ldots, d-1\} \tag{11}$$

Define the "cumulative" contraction factor $R_d$ as follows:

$$R_d = \prod_{t=0}^{d-1} r_t \in (0, 1). \tag{12}$$

This result should be read as a capacity-style impossibility statement for finite-dimensional persistent state under noisy readout, not as an exact performance prediction for the benchmark task. Its role is to formalize a basic limitation: as blank dynamics contract state differences over time, recoverable information about a past write must eventually degrade unless additional signal is injected.

Before providing the theorem, we make one additional remark about Assumption (C3).

*Remark* 4.3. **(Uniform contraction as a natural follow-up to (2) and (3)):** If we let the assumptions of Theorem 1 hold and take the same input $u = v$, then $\sigma(\|u - v\|) = 0$ and (3) becomes

$$W_{t+1}(f_t(x, u), f_t(y, u)) \leq \lambda W_t(x, y). \tag{13}$$

From (2),

$$c_1 \|f_t(x, u), f_t(y, u)\|_2^2 \leq W_{t+1}(f_t(x, u), f_t(y, u)) \tag{14a}$$

---

[2]We abuse the notation and omit the input $u_t$ for brevity.

$$W_t(x, y) \leq c_2 \|x - y\|_2^2. \tag{14b}$$

Combining (14a) and (14b) with (13):

$$c_1 \|f_t(x, u), f_t(y, u)\|_2^2 \leq \lambda W_t(x, y) \leq \lambda c_2 \|x - y\|_2^2.$$

So,

$$\|f_t(x, u), f_t(y, u)\|_2^2 \leq \sqrt{\frac{\lambda c_2}{c_1}} \|x - y\|_2^2, \qquad \forall\, x, y.$$

In the above, if we set $r_t = \sqrt{\frac{\lambda c_2}{c_1}}$ for all $t$, then since $\lambda c_2 < c_1$ by the assumptions of Theorem 1, we get the uniform contraction as stated in (C3). In (C3), we generalize this notion to allow for a time-varying step-wise contraction factor.

With the setup and assumptions in place, we provide the following theorem.

*Theorem 2.* **(Information-limited recall scales with $m$ and decays with blank contraction):** Under Assumptions (C1)–(C3) the following holds:

(I) **(Mutual information bound)**

$$I(H; Y) \leq \frac{m}{2} \log\left(1 + \frac{PR_d^2}{\sigma^2}\right). \tag{15}$$

(II) **(Error lower bound)** For any decoder $\hat{H} = \mathrm{Dec}(Y)$,

$$\mathbb{P}(\hat{H} \neq H) \geq 1 - \frac{I(H; Y) + \log 2}{\log M} \geq 1 - \frac{\frac{m}{2} \log\left(1 + \frac{PR_d^2}{\sigma^2}\right) + \log 2}{\log M}. \tag{16}$$

Equivalently to achieve $\mathbb{P}(\hat{H} \neq H) \leq \varepsilon$, it is necessary that

$$m \geq \frac{2\left((1 - \varepsilon) \log M - \log 2\right)}{\log\left(1 + \frac{PR_d^2}{\sigma^2}\right)}. \tag{17}$$

For a proof, see Appendix.

## 5 Experiments

In this section, we discuss experiments that distinctively fall into two different genres, *viz.* (i) validation and suitability of GSNs in doing dynamic link prediction and (ii) validation of capacity/recall theory. Before presenting the results, it is useful to distinguish two deployment regimes. Some temporal graph architectures encode an explicit history window at query time, which can be effective when moderate latency is acceptable. GSN instead targets settings where prediction is made from a persistent per-node state that is carried forward online, so the cost of the current forward pass does not grow with the amount of accumulated history.

Having set the paradigm, we now present the metrics for GSNs. We start with dynamic link prediction.

All experiments are carried out on a 24 GiB NVIDIA-RTX-4090 GPU.

A practical detail of all reported experiments is that training and evaluation are performed with bucketed updates rather than fully event-by-event streaming updates. This choice is not just an implementation convenience: bucket size controls a speed-fidelity tradeoff. Smaller buckets preserve finer temporal order and, in our preliminary checks on selected datasets, improved predictive metrics, but they require many more state commits and forward passes. Larger buckets reduce compute by amortizing updates over more events, at the cost of coarser temporal resolution.

### 5.1 Inference Cost vs. History Length

Figure 2 empirically validates the constant-time inference claim above. We measure inference latency and profile FLOPs for DyGMamba and GSN as a function of the history context length $L$ on a synthetic graph matching the Wikipedia dataset dimensions (9,228 nodes, batch size 200, same hidden dimension 256 for both models).

**Latency.** DyGMamba's latency grows linearly with $L$ (plus a base/lookup overhead). In particular, each inference call retrieves and re-encodes the last $L$ temporal neighbors for every node in the batch via its node-level and time-level SSM, so compute scales as $O(L/p)$ where $p$ is the patch size. GSN's latency is flat across all $L$: past history is compressed into the state table *online* (during the commit step), so the current-bucket forward pass also includes the *commit-step* and the # of active nodes in the snapshot. DyGMamba's latency continues to grow without bound as $L$ increases, whereas GSN's per-step cost is bounded by the chosen bucket size. This makes bucket size an explicit deployment knob, in that larger buckets reduce the number of commits and forward passes, improving throughput, while smaller buckets increase update frequency and temporal fidelity.

**FLOPs.** Profiled FLOPs confirm the scaling argument independently of hardware. At $L = 512$, DyGMamba requires $262\times$ more FLOPs than GSN. GSN's FLOPs are constant at 243 M across all $L$. DyGMamba FLOPs are measured via `torch.profiler with_flops=True`) and constitute a lower bound, since the `mamba_ssm` selective-scan CUDA kernel is not registered in PyTorch's op-level flop counter; the true gap is therefore larger than reported. GSN FLOPs are measured via `tf.compat.v1.profiler` on a frozen concrete function and represent a complete count.

### 5.2 GSNs for Dynamic Link Prediction

We train GSNs on seven different dynamic link prediction datasets: Wikipedia (**?**), MOOC (**?**), Enron (Shetty & Adibi, 2004), UCI (Panzarasa et al., 2009), US Legis (Poursafaei et al., 2022), Can. Parl (Poursafaei et al., 2022), and Contact (Poursafaei et al., 2022). The results are summarized in Table 1 and Table 2 respectively.

**Protocol and evaluation.** The training process is detailed in Algorithm 1. We follow the standard temporal link prediction protocol used by prior work on these datasets. Each interaction $(u, v, t, \text{attr})$ is processed in timestamp order, and the model is evaluated by scoring the true edge against sampled negatives. We report results under both the commonly used negative sampling regimes shown in the tables (rnd/ind). All evaluations were carried out at bucket sizes specified in Table 1. The benchmark results should be interpreted at a particular bucketed operating point rather than as the intrinsic limit of the architecture under strict streaming updates. In our setup, bucket size controls both computational cost and temporal fidelity: reducing it yields more frequent state updates and better temporal resolution, but can increase runtime dramatically; increasing it reduces update frequency and compute, but aggregates events more coarsely and can degrade predictive quality. In preliminary runs on selected datasets, we observed that reducing bucket size improved the reported metrics, but fully streaming training with bucket size 1 was computationally prohibitive for the larger benchmarks. To make this tradeoff concrete, Table 3 reports a bucket-size sweep on Contact dataset (random negative sampling), showing a clear monotone decrease in AP and AUC-ROC as bucket size increases.

**Baselines.** We compare against representative memory-based and attention-based temporal graph models (e.g., JODIE, DyRep, TGAT, TGN, CAWN, GraphMixer, DyGFormer, DyGMamba), and also against recent sequence-modeling approaches (DyGMamba). Where available, we include strong non-neural heuristics (EdgeBank) and contrastive pretraining baselines (TCL), using the benchmark's reported mean $\pm$ std across runs.

For dynamic link prediction, Average Precision (AP) is the more task-relevant metric: it directly measures whether the true interaction is ranked first among candidates, which corresponds to the actual deployment objective. AUC-ROC measures global pairwise discrimination and is a useful diagnostic but does not directly reflect ranking quality at the top of the candidate list. Because the experiments are bucketed, these metrics

also reflect the chosen temporal granularity: coarser bucketing can blur short-range event order and thereby change both ranking and discrimination quality.

Under AP–the ranking-oriented metric most directly aligned with our evaluation protocol–GSNs are competitive across the seven datasets, with the clearest gains appearing in several inductive settings. The AUC-ROC results are more mixed: on some datasets GSN remains competitive, but on others it trails stronger baselines by a substantial margin. We therefore interpret the benchmark results as showing that persistent node-state modeling can support strong top-of-list ranking performance, especially under inductive shift. Enhancing broader discriminative calibration remains an open direction of work for the current backbone.

Table 1: Average Precision (AP, %) for **transductive** dynamic link prediction with random (rnd) and inductive (ind) negative sampling strategies. APs are reported as mean ± std. AP for all other models are taken from Ding et al. (2025). All results are reported under the bucket sizes used in our practical training/evaluation protocol; they therefore reflect a compute-temporal-resolution operating point rather than strict event-by-event streaming performance.

| NSS | Dataset | JODIE | DyRep | TGAT | TGN | CAWN | EdgeBank | TCL | GraphMixer | DyGFormer | DyGMamba | GSN (bucket size) |
|---|---|---|---|---|---|---|---|---|---|---|---|---|
| rnd | Wikipedia | 96.50 ± 0.14 | 94.86 ± 0.06 | 96.94 ± 0.06 | 98.45 ± 0.06 | 98.76 ± 0.03 | 90.37 ± 0.00 | 96.47 ± 0.04 | 97.25 ± 0.04 | 99.03 ± 0.04 | **99.08 ± 0.03** | 97.05 ± 0.08 (128) |
| | MOOC | 80.23 ± 2.44 | 81.97 ± 0.49 | 85.84 ± 0.15 | 89.15 ± 1.60 | 80.15 ± 0.25 | 57.97 ± 0.00 | 82.38 ± 0.24 | 82.78 ± 0.15 | 87.52 ± 0.49 | 90.25 ± 0.09 | **91.88 ± 0.12** (128) |
| | Enron | 84.77 ± 0.30 | 82.38 ± 3.36 | 71.12 ± 0.97 | 86.53 ± 1.11 | 89.56 ± 0.09 | 83.53 ± 0.00 | 79.70 ± 0.71 | 82.25 ± 0.16 | 92.47 ± 0.12 | **93.14 ± 0.08** | 91.04 ± 0.21 (256) |
| | UCI | 89.43 ± 1.09 | 65.14 ± 2.30 | 79.63 ± 0.70 | 92.34 ± 1.04 | 95.18 ± 0.06 | 76.20 ± 0.00 | 89.57 ± 1.63 | 93.25 ± 0.57 | 95.79 ± 0.17 | **96.14 ± 0.14** | 95.78 ± 0.61 (384) |
| | Can.Parl. | 69.26 ± 0.31 | 66.54 ± 2.76 | 70.73 ± 0.72 | 70.88 ± 2.34 | 69.82 ± 2.34 | 64.55 ± 0.00 | 68.67 ± 2.67 | 77.04 ± 0.46 | 97.36 ± 0.45 | **98.20 ± 0.52** | 97.61 ± 1.15 (256) |
| | USLegis | 75.05 ± 1.52 | 75.34 ± 0.39 | 68.52 ± 3.16 | 75.99 ± 0.58 | 70.58 ± 0.48 | 58.39 ± 0.00 | 69.59 ± 0.48 | 70.74 ± 1.02 | 71.11 ± 0.59 | 73.66 ± 1.13 | **89.21 ± 0.88** (256) |
| | Contact | 95.31 ± 1.33 | 95.98 ± 0.15 | 96.28 ± 0.09 | 96.89 ± 0.56 | 90.26 ± 0.28 | 92.58 ± 0.00 | 92.44 ± 0.12 | 91.44 ± 0.03 | 98.29 ± 0.01 | **98.38 ± 0.01** | 97.90 ± 0.03 (1536) |
| ind | Wikipedia | 75.65 ± 0.79 | 70.21 ± 1.58 | 87.00 ± 0.16 | 85.62 ± 0.44 | 74.06 ± 2.62 | 80.63 ± 0.00 | 86.76 ± 0.72 | 88.59 ± 0.17 | 79.29 ± 5.38 | 87.06 ± 0.86 | **90.78 ± 0.78** (128) |
| | MOOC | 65.23 ± 2.19 | 61.66 ± 0.95 | 75.95 ± 0.64 | 77.50 ± 2.91 | 73.51 ± 0.94 | 49.43 ± 0.00 | 74.65 ± 0.54 | 74.27 ± 0.92 | 81.24 ± 0.69 | 81.19 ± 2.02 | **81.31 ± 0.12** (128) |
| | Enron | 68.96 ± 0.98 | 67.79 ± 1.53 | 63.94 ± 1.36 | 70.89 ± 2.72 | 75.15 ± 0.58 | 73.89 ± 0.00 | 71.29 ± 0.32 | 75.01 ± 0.79 | 77.41 ± 0.89 | 77.46 ± 0.90 | **84.31 ± 0.95** (256) |
| | UCI | 65.99 ± 1.40 | 54.79 ± 1.76 | 68.67 ± 0.84 | 70.94 ± 0.71 | 64.61 ± 0.48 | 57.43 ± 0.00 | 76.01 ± 1.11 | 80.10 ± 0.51 | 72.25 ± 1.71 | 77.75 ± 1.56 | **86.49 ± 0.17** (384) |
| | Can.Parl. | 48.42 ± 0.66 | 58.61 ± 0.86 | 68.82 ± 1.21 | 65.34 ± 2.87 | 67.75 ± 1.00 | 62.16 ± 0.00 | 65.85 ± 1.75 | 69.48 ± 0.63 | 95.44 ± 0.57 | **97.29 ± 0.96** | 96.94 ± 0.98 (256) |
| | USLegis | 50.27 ± 5.13 | 83.44 ± 1.16 | 61.91 ± 5.82 | 67.57 ± 6.47 | 65.81 ± 8.52 | 64.74 ± 0.00 | 78.15 ± 3.34 | 79.63 ± 0.84 | 81.25 ± 3.62 | 85.61 ± 1.66 | **85.78 ± 0.24** (256) |
| | Contact | 93.43 ± 1.78 | 94.18 ± 0.10 | 94.35 ± 0.48 | 90.18 ± 3.28 | 89.31 ± 0.27 | 85.00 ± 0.00 | 91.35 ± 0.21 | 90.87 ± 0.35 | 94.75 ± 0.28 | 94.63 ± 0.06 | **95.43 ± 0.09** (1536) |

Table 2: AUC-ROC (%) for **transductive** dynamic link prediction with random (rnd) and inductive (ind) negative sampling strategies. AUC-ROCs are reported as mean ± std. AUC-ROCs for all other models are taken from Ding et al. (2025). As in Table 1, these numbers are obtained under bucketed processing with bucket size same as Table 1; so the reported AUC-ROC values should be interpreted together with the underlying bucket-size tradeoff between temporal fidelity and compute.

| NSS | Dataset | JODIE | DyRep | TGAT | TGN | CAWN | EdgeBank | TCL | GraphMixer | FreeDyG | DyGFormer | DyGMamba | GSN |
|---|---|---|---|---|---|---|---|---|---|---|---|---|---|
| rnd | Wikipedia | 96.33 ± 0.07 | 94.37 ± 0.09 | 96.67 ± 0.07 | 98.37 ± 0.07 | 98.54 ± 0.04 | 90.78 ± 0.00 | 95.84 ± 0.18 | 96.92 ± 0.03 | **99.41 ± 0.01** | 98.91 ± 0.02 | 99.01 ± 0.09 | 94.14 ± 0.02 |
| | MOOC | 83.81 ± 2.09 | 85.08 ± 0.58 | 87.11 ± 0.19 | **91.21 ± 1.15** | 80.38 ± 0.26 | 60.86 ± 0.00 | 83.12 ± 0.18 | 84.01 ± 0.17 | 89.93 ± 0.35 | 87.91 ± 0.58 | 91.01 ± 0.14 | 89.26 ± 0.67 |
| | Enron | 87.96 ± 0.52 | 84.89 ± 3.00 | 68.89 ± 1.10 | 88.32 ± 0.99 | 90.45 ± 0.14 | 87.05 ± 0.00 | 75.74 ± 0.72 | 84.38 ± 0.21 | **94.01 ± 0.11** | 93.33 ± 0.13 | 93.05 ± 0.17 | 87.26 ± 0.18 |
| | UCI | 90.44 ± 0.49 | 68.77 ± 2.34 | 78.53 ± 0.74 | 92.03 ± 1.13 | 93.87 ± 0.08 | 77.30 ± 0.00 | 87.82 ± 1.36 | 91.81 ± 0.67 | 95.00 ± 0.21 | 94.49 ± 0.26 | **95.32 ± 0.18** | 93.41 ± 0.88 |
| | Can.Parl. | 78.21 ± 0.23 | 73.35 ± 3.67 | 75.69 ± 0.78 | 76.99 ± 1.80 | 75.70 ± 3.27 | 64.14 ± 0.00 | 72.46 ± 3.23 | 83.17 ± 0.53 | N/A | 97.76 ± 0.41 | **98.67 ± 0.29** | 94.35 ± 1.14 |
| | USLegis | 82.85 ± 1.07 | 82.28 ± 0.32 | 75.84 ± 1.99 | **83.34 ± 0.43** | 77.16 ± 0.39 | 62.57 ± 0.00 | 76.27 ± 0.63 | 76.96 ± 0.79 | N/A | 77.90 ± 0.58 | 78.19 ± 0.64 | 79.71 ± 0.82 |
| | Contact | 96.66 ± 0.89 | 96.48 ± 0.14 | 96.95 ± 0.08 | 97.54 ± 0.35 | 89.99 ± 0.34 | 94.34 ± 0.00 | 94.15 ± 0.09 | 93.94 ± 0.02 | N/A | 98.53 ± 0.01 | **98.59 ± 0.00** | 95.80 ± 0.10 |
| ind | Wikipedia | 70.96 ± 0.78 | 67.36 ± 0.96 | 81.93 ± 0.22 | 80.97 ± 0.31 | 70.95 ± 0.95 | 81.73 ± 0.00 | 82.19 ± 0.48 | 84.28 ± 0.30 | 82.74 ± 0.32 | 75.09 ± 3.70 | 82.30 ± 1.81 | **87.83 ± 1.31** |
| | MOOC | 66.63 ± 2.30 | 63.26 ± 1.01 | 73.18 ± 0.33 | 77.44 ± 2.86 | 70.32 ± 1.43 | 48.18 ± 0.00 | 70.36 ± 0.37 | 72.45 ± 0.72 | 78.47 ± 0.94 | 80.76 ± 0.76 | **82.05 ± 1.38** | 78.33 ± 1.76 |
| | Enron | 70.92 ± 1.05 | 68.73 ± 1.34 | 60.45 ± 2.12 | 71.34 ± 2.46 | 75.17 ± 0.50 | 75.00 ± 0.00 | 67.64 ± 0.86 | 71.53 ± 0.85 | 77.27 ± 0.61 | 74.07 ± 0.64 | 74.49 ± 0.48 | **80.38 ± 0.80** |
| | UCI | 64.14 ± 1.26 | 54.25 ± 2.01 | 60.80 ± 1.01 | 64.11 ± 1.04 | 58.06 ± 0.26 | 58.03 ± 0.00 | 70.05 ± 1.86 | 74.59 ± 0.74 | 75.39 ± 0.57 | 65.96 ± 1.18 | 73.23 ± 1.03 | **78.10 ± 0.76** |
| | Can.Parl. | 52.88 ± 0.80 | 63.53 ± 0.65 | 72.47 ± 1.18 | 69.57 ± 2.81 | 72.93 ± 1.78 | 61.41 ± 0.00 | 69.47 ± 2.12 | 70.52 ± 0.94 | N/A | 96.70 ± 0.59 | **97.30 ± 0.97** | 90.09 ± 0.91 |
| | USLegis | 59.05 ± 5.52 | 89.44 ± 0.71 | 71.62 ± 5.42 | 78.12 ± 4.46 | 76.45 ± 7.02 | 68.66 ± 0.00 | 82.54 ± 3.91 | 84.22 ± 0.91 | N/A | 87.96 ± 1.80 | **90.28 ± 0.50** | 87.84 ± 0.31 |
| | Contact | 94.47 ± 1.08 | 94.23 ± 0.18 | 94.10 ± 0.41 | 91.64 ± 0.72 | 87.68 ± 0.24 | 85.87 ± 0.00 | 91.23 ± 0.19 | 90.96 ± 0.27 | N/A | 95.01 ± 0.15 | **95.08 ± 0.01** | 90.70 ± 0.14 |

Table 3: AP and AUC-ROC vs bucket size on Contact dataset (random sampling) on a single run.

| | Bucket sizes | | | |
|---|---|---|---|---|
| | 1536 | 2048 | 3072 | 4096 |
| AP (%) | 97.88 | 96.10 | 95.25 | 93.28 |
| AUC-ROC (%) | 95.91 | 94.23 | 92.71 | 91.01 |

Table 4: Persistence ablation on Wikipedia and USLegis. We disable carried state by resetting the state table to zero after each forward pass, while keeping the architecture, bucket size, and training setup fixed. AP/AUC-ROC (mean ± std, %) are reported over 3 runs for random and inductive negative sampling.

| Dataset/NSS | rnd | | ind | |
|---|---|---|---|---|
| | AP | AUC-ROC | AP | AUC-ROC |
| Wikipedia | 78.60 ± 0.44 | 75.19 ± 1.80 | 63.36 ± 0.46 | 61.60 ± 1.76 |
| USLegis | 70.09 ± 0.07 | 62.19 ± 1.12 | 74.81 ± 0.30 | 60.08 ± 0.98 |

**Persistence ablation.** To test whether GSN benefits from carrying node state across time, we disable state persistence by resetting the state table to zero after each forward pass, while keeping the rest of the architecture, training setup, and bucket size fixed. As shown in Table 4, removing state carry-over causes substantial degradation on both Wikipedia and USLegis under random and inductive negative sampling. These results indicate that GSN's performance is not explained solely by its within-bucket encoder or readout: carrying node state forward over time contributes materially to predictive quality.

Table 5: Updater ablation: GSN with the selective SSM updater replaced by a GRU, while keeping the persistent state table and EMA commit mechanism fixed. AP/AUC-ROC (mean ± std, %) are reported over 3 runs for random and inductive negative sampling.

| Dataset/NSS | rnd | | ind | |
|---|---|---|---|---|
| | AP | AUC-ROC | AP | AUC-ROC |
| Wikipedia | 88.15 ± 0.12 | 76.49 ± 1.31 | 73.49 ± 0.18 | 66.89 ± 2.89 |
| USLegis | 76.46 ± 0.47 | 65.87 ± 2.08 | 69.78 ± 0.26 | 59.21 ± 1.14 |

**Updater ablation.** To isolate the contribution of the selective SSM updater, we replace the SSM backbone in GSN with a GRU while keeping the persistent state table, read–update–commit protocol, and EMA commit rule and the bucket size unchanged (128). As shown in Table 5, this replacement leads to a substantial drop on Wikipedia/USLegis under both random and inductive negative sampling. This suggests that the empirical performance of GSN is not explained solely by persistent node state and EMA-based commitment; the selective SSM update itself contributes materially.

Taken together, Tables 5 and 4 indicate that both ingredients matter: persistent carried state contributes materially, and conditional on having such persistence, the selective SSM updater also provides additional gains.

**Discussion.** Across the seven datasets, GSNs are competitive with recent temporal graph baselines under Average Precision (AP). The benchmark picture is therefore asymmetric. Under AP, GSN often performs strongly, and in several inductive settings it improves materially over prior baselines. Under AUC-ROC, however, performance is inconsistent and in some cases clearly worse than stronger sequence or transformer-style competitors. We attribute the lower AUC-ROC primarily to the bucketed training/evaluation protocol. Because node states are committed only at bucket boundaries, the model does not refresh its state after every event within a bucket. This compresses intra-bucket temporal order and short-range recency information. As a result, candidate edges that differ mainly in fine temporal context become harder to separate, which disproportionately harms AUC-ROC, a global pairwise discrimination metric. Average Precision is affected less severely because it depends mainly on whether the truly relevant interaction is placed near the top of the ranked list. The monotone drop in both AP and AUC-ROC with increasing bucket size in Table 3 supports this temporal-resolution explanation. Our takeaway is not that GSN is uniformly superior, but that persistent *bucketed* node-state modeling is a viable and sometimes advantageous design point whose current strengths lie more in top-rank retrieval than in globally calibrated discrimination. In the next section we answer a more targeted question – *what memory is the model actually retaining, and how does it decay with delay?* We answer that question next using controlled write–wait–read probes that directly test the capacity/recall theory.

### 5.3 Synthetic capacity–recall protocol

This protocol is not intended to reproduce the benchmark task directly; rather, it isolates the core object that the architecture introduces–the persistent node state–and asks whether information intentionally written into that state remains decodable after controlled delay. Each episode consists of a single *write bucket* followed by $d$ *blank buckets*. We randomly choose 256 write nodes. Each written node is assigned a class label from

$$H \in \{1, 2, \ldots, 32\}$$

encoded by a one-hot injection into the node features (node feature dimension 64). After $d$ blank commits, a decoder predicts the per-node class labels from the persistent states of the written nodes.

A blank commit uses the same trained model update as a non-blank commit. Concretely, for a node state $s_t \in \mathbb{R}^m$ the committed update is

$$s_{t+1} = (1 - \alpha)s_t + \alpha \, F_t(s_t), \tag{18}$$

where $m = 64$ and $\alpha \in \{0.05, 0.1, 0.2, 0.4, 1.0\}$ is swept at evaluation time. The map $F_t$ is induced by the *blank bucket context* at step $t$ (graph structure and time features). In the main influence experiments reported below, the blank update includes a small nonzero state-noise scale (0.05), which serves as a persistent stochastic perturbation during blank-bucket commits.

**Time-varying blank schedule.** To construct time-variation while keeping blank buckets uninformative about $H$, each blank step follows a two-regime block schedule with period 5:

$$z_t = \left\lfloor \frac{t}{5} \right\rfloor \mod 2. \tag{19}$$

Thus the blank dynamics alternate between 5 short-regime steps and 5 long-regime steps. Regime $z_t$ controls (i) the effective time-gap presented to the model (bucket size 1 vs. 100, implemented through the timestamp deltas), and (ii) a small regime token injected into node features that is independent of $H$. The blank graphs themselves are sampled independently of $H$.

**Blank-shift control.** As a robustness check, we also evaluate the same trained model on a shifted blank distribution consisting of sparser, undirected graphs with self-loops. This control leaves the write bucket unchanged and only perturbs the blank bucket distribution.

**Role of $\alpha$ in recall.** Figure 3 shows empirical write-vs-zero-write relative influence decay across all five commit rates on a single axes. The ordering is monotone: lower $\alpha$ consistently produces slower decay, while higher $\alpha$ produces faster initial forgetting but saturates at a lower noise floor – supporting the interpretation of $\alpha$ as a practical retention control parameter within the studied backbone and blank-bucket setting. The Theorem-1-style bound tracks each curve closely, connecting the observed behavior to the contraction-with-disturbance analysis as described earlier.

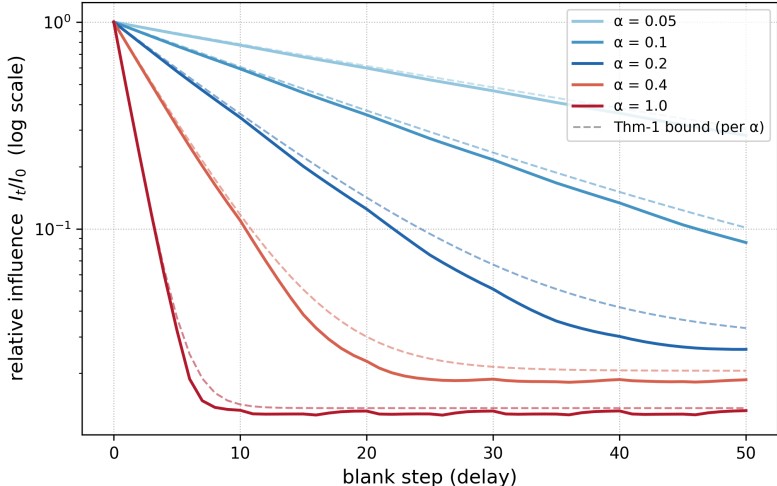

Figure 3: Empirical write-vs-zero-write relative influence decay (solid) and Theorem-1-style contraction-with-disturbance bound (dashed) for each commit rate $\alpha$ under time-varying blank buckets with nonzero state noise. Lower $\alpha$ slows the decay rate while higher $\alpha$ raises the floor, supporting the interpretation of $\alpha$ as a practical retention dial in this setting.

## 5.4 Assumption validation

Here we validate our main assumption, i.e., (3). We choose $W_t(x, y) = \|x - y\|^2$, so that $c_1 = c_2 = 1$ automatically from (2) and hence (2) need not be validated separately whereas (3) requires the contraction ratio $\lambda$ to be less than or equal to 1.0. Define the per-step contraction empirical ratio

$$\lambda_t := \max_{x \neq y, \; u, \; v} \frac{W_{t+1}(f_t(x, u), f_t(y, v)) - \sigma(\|u - v\|)}{W_t(x, y)},$$

where the maximum is taken over all sampled trajectory pairs at blank step $t$ and (3) requires $\lambda_t < 1$ for all $t$. Re-arranging,

$$1 - \frac{W_{t+1}(f_t(x, u), f_t(y, v)) - \sigma(\|u - v\|)}{W_t(x, y)} \geq 1 - \lambda_t.$$

We estimate $\lambda$ using the above equation as

$$\lambda := \max_t \lambda_t. \tag{20}$$

Given this definition, the empirical question is whether the resulting stepwise ratios remain below 1 on the sampled domain for the trained model. Figure 4 plots the min, mean, 95-th percentile and max of the LHS over all sampled pairs at each blank step together with the per-step threshold $1 - \lambda_t$. All four statistics remain above (or equal to) the threshold at every step showing that the sampled trajectories are consistent with (3) on the evaluated domain.

A secondary but important observation follows directly from the plot: since $1 - \max_t \lambda_t > 0$, we have $\max_t \lambda_t < 1$. Thus (20) yields a single time invariant constant satisfying $\lambda \in (0, 1)$, which is precisely the requirement of Theorem 1. In this experiment, we observe $\lambda \approx 0.2$.

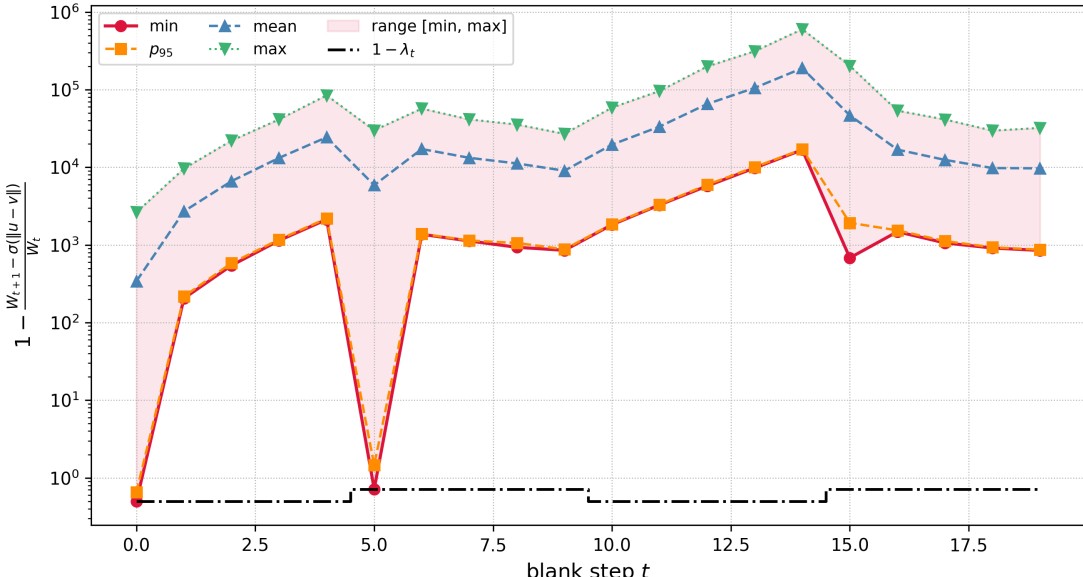

Figure 4: (Red) minimum LHS over all sample paths (trajectories) lie above threshold $1 - \lambda_t$, (Orange) 95-th percentile of all trajectories lie above threshold, (Blue) mean trajectory lies above threshold and (Green) maximum lies above threshold (Black dotted) threshold $1 - \lambda_t$.

### 5.5 Influence measurement

We measure influence by running two counterfactual trajectories that share the same later blank-bucket sequence but differ in the write bucket. Let $x_t$ and $y_t$ denote the resulting persistent states of the written nodes after $t$ blank commits. $x_t$ is initialized from the actual write graph, whereas $y_t$ is initialized from a zero-write baseline. Define the (un-normalized) influence as

$$I_t := \mathbb{E}\big[\|x_t - y_t\|_2\big], \tag{21}$$

and the *relative influence ratio* $I_t/I_0$, which equals 1 at $t = 0$. Also, define the influence *surrogate* to be

$$J_t := \mathbb{E}\big[\|x_t - y_t\|_2^2\big]. \tag{22}$$

Then by Jensen's inequality

$$I_t \leq \sqrt{J_t}.$$

In addition to evaluating influence at a sparse set of delays $d \in \{0, 1, 2, 5, 10, 20, 30, 50\}$, we also compute the *stepwise* influence curve for all blank steps $t \in \{0, 1, \ldots, 50\}$, along with the per-step log-slope

$$\Delta \log I_t := \log(I_{t+1}) - \log(I_t). \tag{23}$$

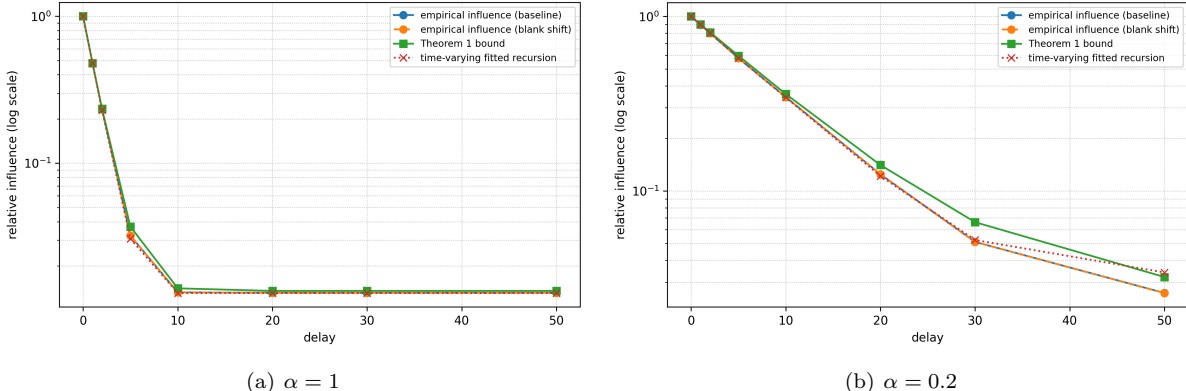

(a) $\alpha = 1$            (b) $\alpha = 0.2$

Figure 5: Write-vs-zero-write influence decay under time-varying blank buckets with nonzero state noise (std. dev = 0.05). Green: Theorem-1-style contraction-with-disturbance bound. Red dotted: fitted time-varying empirical recursion. The high-$\alpha$ panel shows a clear nonzero floor, while the low-$\alpha$ panel shows slower near-exponential decay.

**Theorem 1-style bound and empirical recursion** Theorem 1 motivates a contraction-with-disturbance upper bound on state differences $\Delta_t := \|x_t - y_t\|^2$. In the main noisy-blank setting, we therefore fit a Theorem-1-style affine upper bound of the form

$$\Delta_{t+1} \le a\Delta_t + b$$

to the measured write-vs-zero-write transitions and roll it forward to obtain a predicted relative influence curve. As a more flexible empirical comparator, we also fit a time-varying recursion at the level of squared influence,

$$I_{t+1}^2 \approx a_{z_t} I_t^2 + b_{z_t}, \tag{24}$$

where $z_t$ denotes the blank-bucket regime. The first model is the theory-guided bound shown in Figure 5, while the second is a purely empirical time-varying fit.

## 5.6 Results

Figure 5 shows write-vs-zero-write influence decay under time-varying blank buckets in the main noisy-blank setting (noise scale = 0.05). For $\alpha = 1$, empirical influence decays rapidly and then saturates at a nonzero floor. A Theorem-1-style contraction-with-disturbance bound tracks this behavior closely across delays. For smaller $\alpha$, the decay is more gradual and remains well described by the same theory-guided bound, while the time-varying fitted recursion provides a slightly tighter empirical overlay.

To make time-variation visible beyond sparse delays, Figure 6 plots the full stepwise influence curve and the per-step log-slope for $\alpha = 0.2$ under the same noisy-blank setting. The log-slope is not constant over time, indicating that the effective forgetting rate changes across blank steps under the block-alternating blank regime schedule.

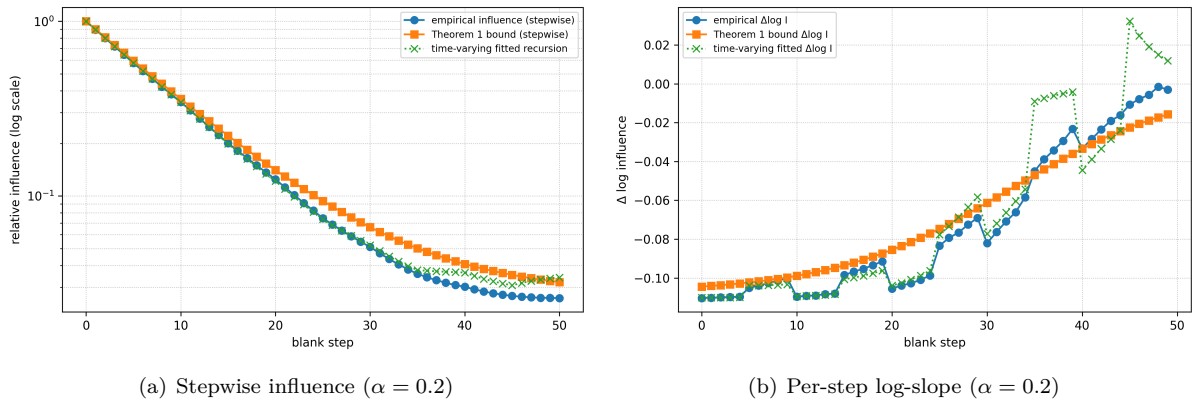

(a) Stepwise influence ($\alpha = 0.2$)  (b) Per-step log-slope ($\alpha = 0.2$)

Figure 6: Stepwise influence and per-step log-slope for $\alpha = 0.2$ under the same noisy-blank setting as Figure 5. A single exponential would yield an approximately constant per-step log-slope, whereas the empirical slope varies over time.

Finally, Figure 7 shows that recall accuracy trends are essentially unchanged under the blank-shift control, suggesting the observed time-variation and non-affine effects are not artifacts of a single narrow blank distribution.

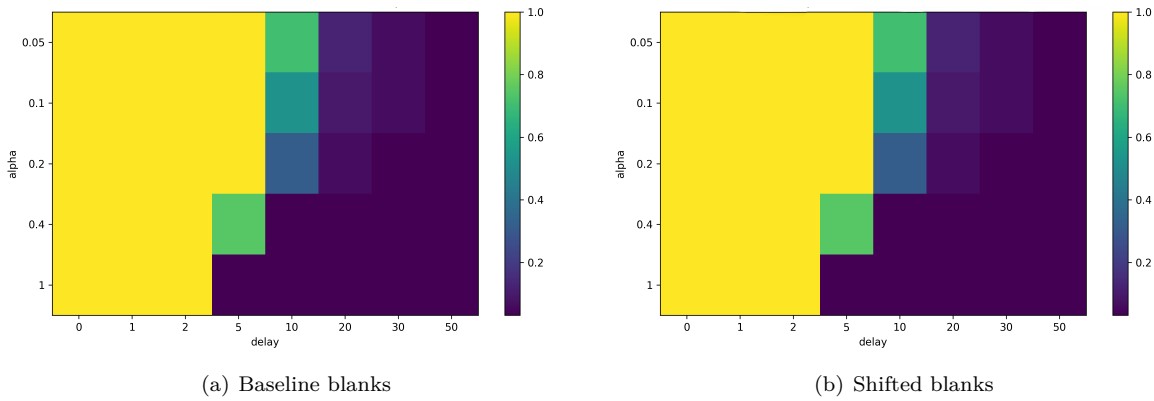

(a) Baseline blanks  (b) Shifted blanks

Figure 7: Recall accuracy heatmaps for baseline blanks and a shifted blank distribution. The qualitative recall behavior is stable under the shift, while time-varying empirical corrections provide a tighter match to the measured decay curves.

# 6   Conclusion

This work proposed **Graph State Networks (GSNs)**: a temporal-graph architecture that treats each node as a **persistent stateful system** stored in an explicit ID-indexed table and updated online via a **time-aware selective state-space block**, with writes committed through an EMA rule controlled by a single **commit rate** $\alpha$. This separation–*how* information is integrated (the backbone update) versus *how aggressively* it is committed ($\alpha$)–yields a simple but useful "retention dial" that is easy to reason about and tune. At the system level, bucket size provides a second practical knob: it trades off temporal fidelity against compute by controlling how often the persistent state table is updated.

On the theory side, we analyzed blank-bucket dynamics under incremental stability assumptions and obtained a contraction-with-disturbance forgetting bound. Complementing this dynamical forgetting result, Theorem 2 provides an information-theoretic capacity perspective: for fixed state dimension and readout

noise, the recoverable message information scales with the state dimension but decays with cumulative blank-bucket contraction. Although we do not isolate this noisy-readout bound in a dedicated experiment here, it formalizes why finite persistent node states cannot support arbitrarily long-horizon exact recall. In the main synthetic setting with small nonzero state noise in the blank update, the measured write-vs-zero-write influence curves are well captured by a Theorem-1-style bound, including the nonzero floor at larger commit rates. Taken together, these results support the interpretation of the commit rate $\alpha$ as a practical retention dial, while also showing that long-horizon behavior depends on the blank dynamics and persistent disturbance, not on $\alpha$ alone. This suggests a practical workflow: choose bucket size to match the available compute/latency budget, then use short-horizon influence measurements to estimate the effective contraction and predict longer-horizon retention at that operating point. Controlled **write–wait–read** probes support this picture: influence curves are close to exponential over the regime that matters for recall, and short-delay fits predict longer-delay behavior well, while deviations at very long delays can be understood as an effective noise floor from time-varying blank operators and backbone nonlinearity. Finally, on standard dynamic link prediction benchmarks, GSNs are competitive under AP and often strong in inductive settings, while AUC-ROC remains less consistent across datasets. Overall, the evidence supports a narrower but concrete claim: GSN's gains are not due to a single component alone–persistent carried state contributes materially, the selective SSM updater adds gains on top of that, and the resulting architecture is competitive primarily under AP rather than uniformly across all metrics.

A few directions are especially promising. First, pushing GSNs closer to strict **event-by-event streaming** (or learning variable bucket sizes) would broaden applicability. Second, $\alpha$ is currently a simple global control knob; learning **adaptive** or **node-specific commit rates** could better trade off retention and adaptation across heterogeneous nodes. Overall, GSNs suggest a useful middle-ground between recomputing history at query time and opaque memory modules: **persistent node memory with a measurable, tunable retention mechanism.** At the same time, the present evidence has clear limits: the theory analyzes a stylized blank-bucket regime rather than the full benchmark dynamics, and the benchmark gains are stronger under AP than under AUC-ROC. Strengthening this bridge between retention theory and end-task performance is the main next step.

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

## A  Proof of Theorem 1

*Proof.* Let $V_t = W_t(x_t, y_t)$. Applying (3) along the two trajectories gives for each $t$,

$$V_{t+1} \leq \lambda V_t + \sigma\big(\|u_t - v_t\|\big).$$

Apply induction:

For $t = 0$: $V_1 \leq \lambda V_0 + \sigma\big(\|u_0 - v_0\|\big)$. Assume for some $t \geq 1$ that

$$V_t \leq \lambda^t V_0 + \sum_{k=0}^{t-1} \lambda^{t-1-k} \sigma\big(\|u_k - v_k\|\big).$$

Then

$$\begin{aligned}
V_{t+1} &\leq \lambda V_t + \sigma\big(\|u_t - v_t\|\big) \\
&\leq \lambda\Big(\lambda^t V_0 + \sum_{k=0}^{t-1} \lambda^{t-1-k} \sigma\big(\|u_k - v_k\|\big)\Big) + \sigma\big(\|u_t - v_t\|\big) \\
&= \lambda^{t+1} V_0 + \sum_{k=0}^{t} \lambda^{t-k} \sigma\big(\|u_k - v_k\|\big)
\end{aligned}$$

which is the same formula for $t \leftarrow t+1$. Therefore, by induction, for all $t$

$$V_t \leq \lambda^t V_0 + \sum_{k=0}^{t-1} \lambda^{t-1-k} \sigma\big(\|u_k - v_k\|\big).$$

Now, using (2): $c_1 \|x_t - y_t\|^2 \leq V_t \leq \lambda^t V_0 + \sum_{k=0}^{t-1} \lambda^{t-1-k} \sigma\big(\|u_k - v_k\|\big)$ and $V_0 \leq c_2 \|x_0 - y_0\|^2$. Therefore,

$$\|x_t - y_t\|^2 \leq \frac{1}{c_1} V_t \leq \frac{c_2}{c_1} \lambda^t \|x_0 - y_0\|^2 + \frac{1}{c_1} \sum_{k=0}^{t-1} \lambda^{t-1-k} \sigma\big(\|u_k - v_k\|\big).$$

Re-writing $\|x_t - y_t\|^2 =: \Delta_t$, we get (4). If $\sigma\big(\|u_k - v_k\|\big) \leq \bar{d}$, the sum is bounded by a geometric series:

$$\sum_{k=0}^{t-1} \lambda^{t-1-k} \sigma\big(\|u_k - v_k\|\big) \leq \bar{d} \cdot \sum_{k=0}^{t-1} \lambda^{t-1-k} = \sum_{j=0}^{t-1} \lambda^j = \bar{d} \cdot \frac{1 - \lambda^t}{1 - \lambda}$$

giving (5) and the $\limsup$ bound. $\qquad \square$

## B  Proof of Theorem 2

*Proof.* Define the composition
$$\Phi_d = f_{d-1} \circ f_{d-2} \circ \cdots \circ f_0,$$
so that $s_d = \Phi_d(s_0)$ by (8). By repeated application of (11) for all $x, y$

$$\|\Phi_d(x) - \Phi_d(y)\|_2 \leq \left(\prod_{t=0}^{d-1} r_t\right) \|x - y\|_2 = R_d \|x - y\|_2.$$

Hence $\Phi_d$ is $R_d$-Lipschitz. Now, let $s_0'$ be an independent copy of $s_0$, i.e.,

$$s_0 \stackrel{d}{=} s_0'.$$

Set $s_d' := \Phi_d(s_0')$. Since $\Phi_d$ is deterministic, $s_d$ and $s_d'$ are i.i.d and in particular

$$s_d \stackrel{d}{=} s_d'.$$

Note that, if $X$ and $X'$ have the same law (i.e., the same underlying distribution), then

$$\mathbb{E}\|X - X'\|_2^2 = 2\mathbb{E}\|X - \mathbb{E}X\|_2^2 = 2\operatorname{tr}(\operatorname{Cov}(X)). \tag{25}$$

Using Lipschitzness of $\Phi_d$,

$$\mathbb{E}\|s_d - s_d'\|_2^2 = \mathbb{E}\|\Phi_d(s_0) - \Phi_d(s_0')\|_2^2 \le R_d^2 \mathbb{E}\|s_0 - s_0'\|_2^2.$$

Now using (25),

$$2\operatorname{tr}(\operatorname{Cov}(s_d)) \le R_d^2 \cdot 2\operatorname{tr}(\operatorname{Cov}(s_0)),$$

hence,

$$\operatorname{tr}(\operatorname{Cov}(s_d)) \le R_d^2 \operatorname{tr}(\operatorname{Cov}(s_0)) \le R_d^2 \cdot mP, \tag{26}$$

where the last inequality follows from Assumption (C2). Now, from (9), $Y = s_d + Z$ with $Z \sim \mathcal{N}(0, \sigma^2 I_m)$ independent. Because $H \to s_d \to Y$ is a Markov chain, the data-processing inequality gives

$$I(H\,;Y) \le I(s_d\,;Y) = I(s_d\,;s_d + Z). \tag{27}$$

Let $K := \operatorname{Cov}(s_d)$, and $h(\cdot)$ denote binary entropy. For any input $X$ with covariance $K$ and independent Gaussian noise $Z \sim \mathcal{N}(0, \sigma^2 I_m)$, the mutual information satisfies

$$
\begin{aligned}
I(X\,;X + Z) &\le h(X + Z) - h(Z) \\
&\le \frac{1}{2}\log\left((2\pi e)^m \det(K + \sigma^2 I_m)\right) - \frac{1}{2}\log\left((2\pi e)^m \det(\sigma^2 I_m)\right),
\end{aligned}
$$

using that for fixed covariance, Gaussian maximizes differential entropy. Thus,

$$I(X\,;X + Z) \le \frac{1}{2}\log\det\left(I_m + \sigma^{-2}K\right).$$

Applying this with $X = s_d$ yields

$$I(s_d\,;s_d + Z) \le \frac{1}{2}\log\det\left(I_m + \sigma^{-2}K\right). \tag{28}$$

Let $\lambda_1, \ldots, \lambda_m \ge 0$ be the eigenvalues of $\sigma^{-2}K$. Then,

$$\det\left(I_m + \sigma^{-2}K\right) = \prod_{i=1}^{m}(1 + \lambda_i).$$

By AM-GM inequality,

$$\prod_{i=1}^{m}(1 + \lambda_i) \le \left(\frac{1}{m}\sum_{i=1}^{m}(1 + \lambda_i)\right)^m = \left(1 + \frac{1}{m}\sum_{i=1}^{m}\lambda_i\right)^m = \left(1 + \frac{\operatorname{tr}(K)}{m\sigma^2}\right)^m. \tag{29}$$

Taking logs and combining (28), (29), and using $\operatorname{tr}(K) \le mPR_d^2$, we get

$$I(s_d\,;s_d + Z) \le \frac{m}{2}\log\left(1 + \frac{PR_d^2}{\sigma^2}\right).$$

Combining this with (27) proves conclusion (I).

Now, let $P_e := \mathbb{P}(\hat{H} \neq H)$. Let $\mathbb{H}$ denote the Shannon entropy. Then, Fano's inequality gives

$$\mathbb{H}(H \mid Y) \leq h(P_e) + P_e \log(M-1) \leq \log 2 + P_e \log M \tag{30}$$

since $h(P_e) \leq \log 2$. Also,

$$I(H\,;\,Y) = \mathbb{H}(H) - \mathbb{H}(H \mid Y) = \log M - \mathbb{H}(H \mid Y). \tag{31}$$

Combining (30) and (31) gives

$$I(H\,;\,Y) \geq \log M - (\log 2 + P_e \log M) = (1 - P_e) \log M - \log 2,$$

so

$$P_e \geq 1 - \frac{I(H\,;\,Y) + \log 2}{\log M}$$

which is the first inequality in (16). The second inequality in (16) follows by applying (15). Rearranging the above gives (17). □

## C GSN Training Algorithm

---

**Algorithm 1** Bucketed training of a Graph State Network (GSN) for temporal link prediction

---

**Require:** Temporal event stream $\mathbf{E}$; cached negatives Neg; bucket size $B_e$ (events per bucket); optimizer Opt; accumulation steps $A$; commit schedule $\alpha(\cdot)$ (or constant $\alpha$); write-penalty weight $\lambda_{\mathrm{wr}}$; scorer head Score($\cdot$).

1: Initialize persistent state table(s) $\mathbf{S}$ (indexed by global node IDs).
2: Initialize gradient accumulators $\{\Delta\theta\} \leftarrow 0$ and counter $c \leftarrow 0$.
3: **for** epoch $= 1, 2, \ldots, E$ **do**
4:     **for** bucket $k$ in chronological order **do**
5:         $(\mathbf{u}, \mathbf{v}^+, \mathbf{t}, \mathbf{v}^-) \leftarrow \text{NextTimeBatchWithNeg}(\mathbf{E}, \text{Neg}, B_e)$
6:         $[t_{\mathrm{start}}^{(k)}, t_{\mathrm{end}}^{(k)}] \leftarrow [\min(\mathbf{t}), \max(\mathbf{t})]$
7:         $G_{\mathrm{prev}}^{(k)} \leftarrow \text{Snapshot}\left(\mathbf{E};\ t < t_{\mathrm{start}}^{(k)}\right)$              ▷ history-only
8:         $G_{\mathrm{end}}^{(k)} \leftarrow \text{Snapshot}\left(\mathbf{E};\ t \le t_{\mathrm{end}}^{(k)}\right)$             ▷ history + bucket
9:         $(\mathbf{P}, \mathbf{s}) \leftarrow \text{BuildCandidates}(\mathbf{u}, \mathbf{v}^+, \mathbf{v}^-)$
10:        $\text{TrainStep}(G_{\mathrm{prev}}^{(k)}, \mathbf{P}, \mathbf{s}, \lambda_{\mathrm{wr}};\ \mathbf{S}, \alpha(\cdot), \text{Opt}, \{\Delta\theta\}, c, A)$
11:        $\text{CommitToEnd}(G_{\mathrm{end}}^{(k)};\ \mathbf{S}, \alpha(\cdot))$         ▷ updates $\mathbf{S}$ for bucket $k{+}1$
12:     **end for**
13: **end for**
14: **function** BuildCandidates($\mathbf{u}, \mathbf{v}^+, \mathbf{v}^-$)
15:     $\mathbf{P} \leftarrow [\,]$,  $\mathbf{s} \leftarrow [\,]$
16:     **for** $i = 1$ to $|\mathbf{u}|$ **do**
17:         $\mathbf{C}_i \leftarrow [\,v_i^+\,] \,\|\, \mathbf{v}_i^-$           ▷ positive first
18:         $\mathbf{s}[i] \leftarrow |\mathbf{C}_i|$
19:         **for** each $v \in \mathbf{C}_i$ **do**
20:             append pair $(u_i, v)$ to $\mathbf{P}$
21:         **end for**
22:     **end for**
23:     **return** $(\mathbf{P}, \mathbf{s})$
24: **end function**
25: **function** TrainStep($G, \mathbf{P}, \mathbf{s}, \lambda_{\mathrm{wr}};\ \mathbf{S}, \alpha(\cdot), \text{Opt}, \{\Delta\theta\}, c, A$)
26:     Let $\{u_i\}_{i=1}^B$ be the source IDs for each query (one per block), inferred from $\mathbf{P}$ and $\mathbf{s}$.
27:     **Read pre-states:**  $\mathbf{s}_{\mathrm{pre}} \leftarrow \mathbf{S}[u_1, \ldots, u_B]$
28:     **Forward + commit on history snapshot:**
29:     Run GSN on $G$ with commit enabled to obtain (i) node embeddings $\mathbf{H}$ for nodes in $G$ and (ii) post-update states $\mathbf{s}_{\mathrm{post}}$ for touched sources.
30:     **Score candidates (post):**  $\ell_{\mathrm{post}} \leftarrow \text{Score}(\mathbf{s}_{\mathrm{post}}, \mathbf{H}, \mathbf{P})$
31:     **Ranking loss:**  $L_{\mathrm{rank}} \leftarrow \text{RankLoss}(\ell_{\mathrm{post}}, \mathbf{s})$         ▷ CE/BCE over each block
32:     **Score-change-normalized write penalty:**
33:     $\Delta\mathbf{s} \leftarrow \mathbf{s}_{\mathrm{post}} - \text{stopgrad}(\mathbf{s}_{\mathrm{pre}})$
34:     $d_s \leftarrow \|\Delta\mathbf{s}\|_2$             ▷ per-source norm
35:     $\ell_{\mathrm{pre}} \leftarrow \text{Score}(\text{stopgrad}(\mathbf{s}_{\mathrm{pre}}), \mathbf{H}, \mathbf{P})$
36:     $d_{\mathrm{score}} \leftarrow \text{BlockMean}(|\ell_{\mathrm{post}} - \ell_{\mathrm{pre}}|, \mathbf{s})$         ▷ mean over each block
37:     $L_{\mathrm{wr}} \leftarrow \lambda_{\mathrm{wr}} \cdot \text{mean}\left(\dfrac{d_s}{\text{stopgrad}(d_{\mathrm{score}}) + \varepsilon}\right)$
38:     $L \leftarrow L_{\mathrm{rank}} + L_{\mathrm{wr}}$
39:     **Backprop + accumulate:**  $g \leftarrow \nabla_\theta L$;  $\Delta\theta \leftarrow \Delta\theta + g$;  $c \leftarrow c + 1$
40:     **if** $c \bmod A = 0$ **then**
41:         Opt.Step($\Delta\theta/A$);  $\Delta\theta \leftarrow 0$
42:     **end if**
43: **end function**
44: **function** CommitToEnd($G_{\mathrm{end}};\ \mathbf{S}, \alpha(\cdot)$)
45:     Run GSN on $G_{\mathrm{end}}$ with commit enabled (no scoring), updating persistent table(s) $\mathbf{S}$
46:     via EMA commit: $\mathbf{S}[i] \leftarrow (1 - \alpha)\mathbf{S}[i] + \alpha\mathbf{s}_i'$ (applied by the model)
47: **end function**

---

