# OpenReview forum: "Graph State Networks (GSNs): Persistent Nodewise Selective State Space Models"
_TMLR — Accepted by TMLR_

### Review · Reviewer_ZAhU · 2026-02-27

**Summary Of Contributions:**

Summary:

Graph State Networks (GSNs) introduce a persistent node-centric architecture for temporal graphs, in which each node maintains a hidden state stored in an explicit ID-indexed table and updated online through a time-aware selective state-space model. Instead of recomputing representations from historical neighborhoods at query time, GSNs adopt a read–update–commit protocol where new state proposals are integrated via an exponential moving average controlled by a commit rate α, serving as an interpretable retention dial. The paper develops a theoretical analysis of blank-bucket dynamics, showing that under an affine approximation the influence of a past event decays geometrically with a contraction factor jointly determined by α and the induced linearized operator  ￼. Empirical results on standard dynamic link prediction benchmarks demonstrate competitive performance, while synthetic write–wait–read probes validate the proposed capacity/recall theory and confirm that short-horizon measurements can accurately predict long-term retention behavior.

Strengths:
1. GSNs provide a clean separation between state integration (selective SSM update) and state persistence (EMA commit), yielding an interpretable and tunable retention mechanism via the commit rate α.
2. The blank-bucket theory offers closed-form geometric decay bounds under an affine approximation, connecting architectural design to measurable memory half-life and contraction dynamics.
3. The model meaningfully integrates selective state-space models (Mamba-style updates) into persistent node memory, extending SSM ideas beyond sequence modeling to temporal graph settings.
4. In addition to competitive dynamic link prediction results, the synthetic write–wait–read probes rigorously validate the theoretical predictions, strengthening the methodological contribution.

Weaknesses:
1. The blank-bucket analysis relies on an affine linearization of the update operator, whereas the fully trained model exhibits significant nonlinearity and time-varying dynamics; therefore, the external validity of the theoretical results in complex dynamic environments still requires more rigorous justification.
2. In practice, blank buckets induce time-varying operators, yet the theory is primarily developed under a time-invariant assumption, lacking stability and convergence guarantees for stochastic or time-varying linear system settings.
3. Although the paper introduces a capacity/recall perspective, it does not provide formal upper bounds related to the state dimension m, nor does it establish information-theoretic limits on compression error or recoverable information.
4. On several datasets, GSN does not consistently outperform the latest SSM-based or Transformer-based models in terms of AUC-ROC, suggesting that there remains substantial room for further engineering optimization and architectural refinement.

**Audience:**

Yes

**Audience Explanation:**

Despite the empirical limitations, the paper introduces an interesting architectural perspective that treats each node as a persistent stateful system with an explicit retention mechanism, which may attract researchers working on temporal graphs, state-space models, and memory-augmented architectures. In particular, the combination of selective state-space updates with an explicit commit-rate “retention dial,” along with the accompanying theoretical analysis of forgetting dynamics, provides conceptual insights that could stimulate further discussion and follow-up work within the TMLR community.

**Claims And Evidence:**

No

**Claims Explanation:**

The claims made in the submission are not fully supported by sufficiently strong empirical evidence. Although the paper presents an interesting architectural design and a well-motivated theoretical analysis, the experimental results are not consistently competitive with recent SSM-based or Transformer-based baselines, particularly in terms of AUC-ROC across multiple datasets. The reported improvements are dataset-dependent and do not demonstrate clear, robust advantages under both transductive and inductive settings. Moreover, while the retention/forgetting theory is elegant, the empirical evaluations do not convincingly show that this theoretical contribution translates into meaningful performance gains in practical dynamic link prediction tasks.

**Requested Changes:**

See the Weaknesses.

---

> ### Author Response · Authors · 2026-03-16
> **Response to Reviewer**
>
> We thank the reviewer for the detailed feedback. Below, we respond to the points raised by the reviewer, clarify the scope of our claims, and describe concrete revisions that address points (1)-(3), while we acknowledge that there remains substantial engineering gap in terms of AUC-ROC as noted in (4). We have revised the manuscript accordingly and have posted a revised version.
>
> (1) Affine linearization vs. nonlinear dynamics.
>
> We had used an affine approximation for our blank-bucket analysis to make forgetting behavior easier to illustrate. We agree with the reviewer, however, that the trained update is nonlinear and that a purely time-invariant affine model will not capture all long-horizon effects. In this revision, this has been replaced by a time-varying non-affine treatment. Specifically, we now analyze forgetting under incremental stability assumptions for time-varying blank-bucket dynamics; Section 4.2 introduces a time-varying discrete-time system $x_{t+1} = f_t(x_t, u_t)$ with an incremental ISS-Lyapunov family $W_t$, and Theorem 1 gives an exponential forgetting bound with a disturbance floor, without relying on a time-invariant model.
>
> (2) Time-varying blank operators and stability guarantees.
>
> We are indebted to the reviewer for this insightful feedback. In particular, we agree that, in realistic streams, blank buckets induce time-varying operators (e.g., different blank graphs and time gaps). The ISS result above is explicitly time-varying and provides a stability/convergence guarantee under standard contraction assumptions. In Section 3.4 we define the committed bucket operator as
> $$
>     T_\alpha(s) = (1-\alpha)s + \alpha F_t(s),
> $$
> with $F_t$ induced by bucket-specific context. Section 5.2 then adds a two-regime block-alternating blank schedule with different time gaps and graph statistics, while keeping blank buckets uninformative about the payload. Section 5.3 measure write-vs-zero-write influence under shared subsequent blank sequences, and Section 5.4 shows both the stepwise influence curve and the per-step log-slope, making the time the time variation visible empirically. The revised experiments also include a blank-shift control showing that recall trends remain qualitatively stable under a shifted blank distribution.
>
> (3) Capacity/recall and dependence on state dimension $m$
>
> We agree with this point. This has now been formalized in Section 4.3. The revised manuscript introduces Theorem 2, which models readout as $Y = s_d + Z$ with Gaussian noise $Z$, and assumes bounded write energy and cumulative blank contraction, and derives (i) a mutual-information upper-bound
> $$
> I(H;Y)\le \frac{m}{2}\log\Big(1+\frac{P R_d^2}{\sigma^2}\Big),
> $$
> followed by (ii) a Fano-style decoding error lower bound and a necessary scaling condition on $m$. This directly addresses the reviewer's concern for an explicit dependence on state dimension and recoverable information.
>
> (4) End-to-end link prediction performance.
>
> We agree with the reviewer’s point that GSN does not consistently exceed the strongest recent SSM/Transformer baselines on AUC-ROC across all datasets. We revised the phrasing to avoid implying universal competitiveness across all metrics and position the empirical results as evidence that persistence plus a controllable commit rule can remain competitive under standard protocols.

---

### Review · Reviewer_syyn · 2026-03-19

**Summary Of Contributions:**

The paper proposes Graph State Networks, a temporal graph model that stores a persistent hidden state for each node in an ID-indexed table. For each bucket, the model reads the current state, applies a time-aware selective state space update, and commits the proposal back with an EMA controlled by commit rate $\alpha$. The paper also develops blank-bucket theory: Theorem 1 gives exponential forgetting under incremental stability assumptions, and Theorem 2 gives an information bound under contractive blank dynamics plus noisy readout. Experiments include dynamic link prediction on seven datasets and synthetic write-wait-read probes intended to study retention.

**Strengths:**

**S1.** The paper makes retention a first-class object and uses a simple commit rule, $\alpha$, that is easy to interpret.

**S2.** The synthetic write-wait-read protocol is a reasonable way to probe forgetting under controlled blank dynamics.

**S3.** The AP results are competitive on several inductive settings, especially Wikipedia, Enron, and UCI.

**Additional Comments:**

**C1.** Section 3.2: the notation $F_\phi$ takes $(s_i; \text{snapshot context}, \Delta t)$ but the earlier signature says $F_\phi : \mathbb{R}^m \times \mathcal{E} \times [0,T] \to \mathbb{R}^m$. The edge set $\mathcal{E}$ and "snapshot context" are used interchangeably.

  **C2.** Baseline naming is inconsistent: DyGMamba in text vs. DyG-Mamba in tables; FreeDyG appears only in Table 2 with no mention in the baselines paragraph.

  **C3.** Figures 2-4 would benefit from larger axis labels and clearer legends.

**Audience:**

Yes

**Audience Explanation:**

Persistent node memory for temporal graphs is an interesting topic, and the attempt to analyze retention rather than only reporting link prediction scores is relevant to TMLR readers.

**Broader Impact Concerns:**

None.

**Claims And Evidence:**

No

**Claims Explanation:**

The main evidence does not fully support the paper's core claims, for the following reasons:

**W1.** Section 3.3 says bucket $k$ is scored on $G_{\text{prev}}^{(k)}$, and the persistent state is committed only after scoring using $G_{\text{end}}^{(k)}$. But Algorithm 1 runs `TrainStep(Gprev, ...)` with “forward + commit” enabled, then runs `CommitToEnd(Gend, ...)` again. Since $S$ persists across buckets, the paper does not make clear whether the first pass mutates the real state table or whether that commit is only virtual.

  **W2.** Section 4 proves results for a contracting time-varying dynamical system under external assumptions, not for the learned GSN update itself. The paper does not show that the selective SSM blank-bucket operator satisfies the incremental ISS or uniform contraction assumptions used in Theorems 1 and 2. The empirical validation is also limited, because Section 5.3 fits a Theorem-1-style recurrence directly on the measured influence trajectories, and Section 6 states that Theorem 2 is not isolated in a dedicated experiment.

  **W3.** The key “retention dial” claim is not isolated end to end. The main $\alpha$ sweep appears only in the synthetic write-wait-read probe at evaluation time, and the paper does not separately show what comes from the EMA commit rule versus the backbone update itself. As written, the experiments show that the full system can exhibit controllable decay in the synthetic setting, but they do not cleanly identify the source of that effect.

  **W4.** The benchmark evidence supports the narrower AP claim better than the broader performance claim. Table 1 is competitive on AP in several settings, especially some inductive cases, but Table 2 is often much weaker on AUC than strong baselines. Given that the conclusion says the method is “broadly competitive” and “particularly strong” in inductive settings, the empirical discussion feels overstated.

  **W5.** Section 5 contrasts GSNs with bidirectional baselines and claims immediate updates with constant-time inference compatibility in streaming settings. But the paper also describes the actual implementation as bucketed, not strict event-by-event streaming, and it provides no runtime, latency, or memory evidence. So the efficiency/online claim is asserted more strongly than it is demonstrated.

**Requested Changes:**

Solve or clarify W1-W5.

---

> ### Author Response · Authors · 2026-03-25
> **Response to Reviewer**
>
> We thank the reviewer for the detailed and constructive feedback. We have revised the manuscript and posted a revised version in which we address each of the weakness below.
>
> **1. *Algorithm 1 commit ambiguity.***
>
> The two-pass structure is intentional and non-redundant. Within bucket $k$, `TrainStep(G_prev)` runs on history strictly before `t_start`, scoring candidate edges without leakage while simultaneously committing the pre-bucket context into $S$. `CommitToEnd(G_end)` then incorporates the bucket-$k$ events themselves, so that the state entering bucket $k{+}1$ reflects all interactions up to $t^{(k)}_{\mathrm{end}}$. These two commits operate on *different* snapshots and serve different purposes; neither is virtual or redundant. This is now clarified in Section 3.3.
>
> **2. *ISS/contraction assumptions not verified for the learned GSN.***
>
> We agree that Theorems 1 and 2 are stated for general systems satisfying the incremental ISS and uniform contraction conditions respectively, and do not prove these hold *a priori* for the learned backbone. Section 5.4 empirically validates the key inequality (3) on the trained model by measuring the per-step contraction ratio $\lambda_t$ over all sampled trajectory pairs and confirming that $\max_t \lambda_t < 1$, which is precisely the requirement of Theorem 1. Remark 4.2 further shows that the uniform contraction in Assumption (C3) follows directly from the incremental ISS constants $(c_1, c_2, \lambda)$, so a single empirical check covers both theorems. We acknowledge that this constitutes post-hoc validation rather than an analytic certificate, and have sharpened the language in Section 4 to state this scope clearly.
>
> **3. *Retention dial not isolated end-to-end.***
>
> We have added a dedicated experiment and figure (Figure 3, Section 5.3) that sweeps all five commit rates $\alpha \in \{0.05, 0.1, 0.2, 0.4, 1.0\}$ on a single axes. The empirical write-vs-zero-write relative influence curves are monotonically ordered by $\alpha$. This isolates $\alpha$ as a direct, quantifiable retention dial independent of backbone dynamics, since the backbone weights are fixed across the sweep and only the commit rate varies.
>
> **4. *Benchmark claims overstated relative to AUC-ROC evidence.***
>
> We have restructured Section 5.2 to establish the metric hierarchy before presenting results. Average Precision (AP) is the primary metric because it directly measures whether the true interaction is ranked first among candidates, which matches the deployment objective. AUC-ROC measures global pairwise discrimination and is a useful diagnostic but does not directly reflect top-of-list ranking quality.
>
> **5. *Streaming/efficiency claim asserted without latency evidence.***
>
> We have added Section 5.1 with an empirical inference-cost comparison against DyGMamba as a function of history context length $L$. Profiled FLOPs show that GSN FLOPs are constant at 243 M for all $L$, while DyGMamba requires $262{\times}$ more FLOPs at $L = 512$. Latency measurements confirm a flat GSN profile vs. an $O(L)$ growing profile for DyGMamba. We also clarify that the current implementation is bucketed rather than strict event-by-event streaming, and that event-level streaming is a limiting case achieved with bucket size 1, as noted in Section 3.

---

### Review · Reviewer_D9fE · 2026-04-02

**Summary Of Contributions:**

The paper introduces Graph State Networks (GSNs), a temporal graph model in which each node maintains a persistent state that is updated online rather than recomputed from historical neighborhoods. The model combines a time-aware selective state-space update with an exponential moving average commit rule, where the commit rate acts as a controllable parameter governing memory retention. On the theoretical side, the authors analyze the evolution of node states under “blank” updates and derive bounds showing geometric decay of past information and limits on recoverable memory due to finite state dimension. Empirically, they propose synthetic write–wait–read probes to study retention dynamics and demonstrate that the model’s behavior aligns with the theoretical predictions. On standard dynamic link prediction benchmarks, GSNs are competitive—particularly under Average Precision and in inductive settings—while offering constant-time inference with respect to history length. The framework also exposes a practical tradeoff between temporal fidelity and computational cost through bucketed processing.

**Audience:**

Yes

**Audience Explanation:**

Yes, researchers working on temporal graphs, dynamic link prediction, and graph representation learning, as the paper proposes a distinct paradigm based on persistent node-level memory. It is also relevant to those studying state-space models and their extension to structured data, as well as researchers interested in efficient modeling of long-range dependencies and controllable memory mechanisms. So for readers already engaged in temporal graph modeling or memory-based architectures it would be of interest.

**Broader Impact Concerns:**

The paper would benefit from briefly discussing how retention mechanisms (e.g., the α parameter) could be used to control or limit information persistence, and whether this introduces risks related to privacy, bias amplification over time, or unintended memorization of historical data.

**Claims And Evidence:**

No

**Claims Explanation:**

**On evaluation protocol and comparability**

* The revised results are obtained under different bucket sizes. Can the authors provide a **strict apples-to-apples comparison**, where GSN and all baselines are evaluated under identical temporal resolution (same bucket size or true event-level streaming)?
* How sensitive are the reported gains to bucket size across all datasets, not just Contact? Can the authors provide **full bucket-size sweeps for each benchmark**?
* Since bucket size directly affects both compute and accuracy, what is the **Pareto frontier (accuracy vs compute)** compared to baselines?

---

**On the role of persistence vs architecture**

* The persistence and updater ablations are limited to two datasets. Do these findings **generalize across all benchmarks**?
* How does performance change as a function of **state dimension**, i.e., is the benefit due to persistence or simply increased capacity?

---

**On theoretical claims and their relevance**

* The theory is derived for blank-bucket dynamics. Can the authors demonstrate that the same **forgetting behavior holds during real task execution**, where informative signals are continuously present?
* Is there any empirical link between the **theoretical retention bounds and downstream task performance** (e.g., does slower decay correlate with better prediction)?
* The stability assumptions are only validated empirically on sampled trajectories. How robust are these assumptions across datasets and training regimes?

---

**On the “retention dial” interpretation**

* The interpretation of α as a retention control is validated in synthetic probes. Does varying α lead to **predictable and consistent changes in real benchmark performance**?
* Can the authors quantify the **tradeoff between retention (α) and predictive accuracy** in real datasets?

---

**On realism of the experimental setting**

* The model is motivated as an online, streaming system, yet experiments use bucketed updates. Can the authors provide results closer to **true event-by-event streaming**, even on smaller datasets?
* How does the method behave under **distribution shift over time**, where older information may become irrelevant?

---

**On efficiency claims**

* The constant-time inference claim assumes fixed bucket size. What is the **end-to-end cost (training + inference)** compared to baselines at matched accuracy?
* Does the cost advantage persist when using **smaller buckets (higher temporal fidelity)**?

**Requested Changes:**

**Critical:** Provide strictly comparable evaluations by fixing the temporal resolution (e.g., same bucket size or true streaming) across GSN and all baselines, or report a clear compute–accuracy Pareto analysis. As it stands, performance claims are confounded by the bucket-size operating point, making it difficult to attribute gains to the model rather than to increased temporal fidelity.

**Critical:** Strengthen the empirical validation of the main claims by expanding ablations across all datasets, including persistence removal, updater replacement, and sensitivity to state dimension and α. This is necessary to demonstrate that the observed effects are robust and not limited to a small subset of benchmarks.

**Important but not critical:** Clarify the relationship between the theoretical analysis and practical performance by providing evidence that the retention/forgetting behavior observed in synthetic probes correlates with downstream task metrics. This would help justify the relevance of the theory beyond the stylized blank-bucket setting.

**Important but not critical:** Provide additional analysis of metric asymmetry (AP vs AUC-ROC), including discussion or experiments showing when and why the method should be preferred, especially in applications requiring calibrated discrimination rather than top-rank retrieval.

**Optional but strengthening:** Include experiments closer to true streaming (even on smaller datasets) and report end-to-end efficiency comparisons at matched accuracy levels. This would better support the claimed advantages in deployment scenarios.

---

> ### Author Response · Authors · 2026-04-05
> **Author Response**
>
> Thank you for the careful and constructive review. We appreciate that you find the topic relevant to TMLR, and we agree that the main issue is to separate what is already supported by the current evidence from what would require a larger follow-up study.
>
> **On comparability and bucket size.**
> We agree that bucket size affects both temporal fidelity and compute. Our revised manuscript is intended to make this dependence explicit rather than implicit. Accordingly, our claim is **not** that GSN has been shown superior under matched event-level temporal resolution for all methods. The narrower claim we make is that, under practical **bucketed** operating points, persistent node-state modeling is competitive under AP, with the strongest gains appearing in several inductive settings, while bucket size exposes a clear compute-fidelity tradeoff. The smaller-bucket reruns and the bucket-size sweep were added precisely to make this operating-point dependence visible.
>
> **On persistence vs. architecture.**
> We agree this distinction matters. Tables 4-5 were added to separate these effects, i.e., removing carried state causes substantial degradation, and replacing the selective SSM updater with a GRU also causes substantial degradation. Our intent is therefore not to argue that the gains come from a single ingredient, but that both persistent carried state and the selective SSM updater contribute materially. We also agree that broader sweeps over state dimension, $\alpha$, and all datasets would be valuable; however, we view those as extensions beyond the scope of the current revision rather than prerequisites for the narrower claim above.
>
> **On theory and practical relevance.**
> The theory is intentionally stated for **blank-bucket dynamics**, not as a theorem for the full downstream benchmark setting. Its role is to analyze the retention mechanism in isolation. The synthetic write-wait-read probes were designed for exactly this purpose: to test whether the learned model exhibits the predicted contraction-with-disturbance behavior in a controlled regime. We therefore view the benchmark results and the retention theory as complementary: the former shows end-task competitiveness, while the latter explains a specific retention mechanism of the persistent state.
>
> **On the "retention dial" interpretation.**
> Our claim here is likewise scoped. We empirically validate $\alpha$ as a retention-control parameter in the controlled synthetic setting studied in the paper. We do **not** claim that varying $\alpha$ alone should produce a simple monotone improvement on every real benchmark, since end-task performance also depends on bucket size, backbone dynamics, and the retention/adaptation tradeoff.
>
> **On streaming and efficiency.**
> We agree that the implementation is bucketed rather than strict event-by-event streaming. Accordingly, the constant-time claim is with respect to **history length** at a fixed bucket size, not a claim of matched-accuracy end-to-end dominance under all temporal resolutions. Figure 2 is intended to support precisely this scoped claim.
>
> Overall, we hope this clarifies the intended scope of the paper: a **bucketed persistent-state architecture with an explicit and measurable retention mechanism**, competitive especially under AP/inductive evaluation, rather than a definitive event-level Pareto comparison against all baselines.

---

### Decision · Action_Editor_pECa · 2026-05-13

**Recommendation:** Accept as is

**Additional Comments:**

This paper introduces Graph State Networks (GSNs), a persistent node-state architecture for temporal graphs where states are maintained in a global table and updated via a content-dependent selective State Space Model (SSM). The model utilizes an explicit "read-update-commit" protocol with an exponential moving average (EMA) commit rate $\alpha$, which acts as a controllable "retention dial" for memory. Theoretically, the work provides geometric forgetting bounds under blank-bucket dynamics, while empirically demonstrating computational efficiency and competitive performance on dynamic link prediction.

The paper initially received mixed reviews. While the reviewers appreciated the idea of the retention mechanism, the "write-wait-read" protocol, and the integration of SSMs into persistent node memory, they also identified two major areas of weakness. The primary concern across all reviews was that the initial claims were not sufficiently supported by evidence. Reviewers requested that the authors narrow their claims and provide new experiments to validate the core contributions.The rebuttal effectively addressed these concerns. The authors clarified the scope of their claims, verified theoretical assumptions (ISS contraction) with explicit experiments, and re-ran benchmarks on several datasets (MOOC, UCI, Can. Parl., and US Legis.) using smaller bucket sizes. Furthermore, they provided clear evidence regarding the accuracy-vs-compute trade-off. Following the rebuttal and the submission of the revised manuscript, two reviewers (D9fE and syyn) expressed satisfaction with the updates and leaned toward acceptance.


The AE has thoroughly reviewed the submission and the discussion. The AE acknowledges that the combination of SSMs and persistent node memory is a meaningful contribution to dynamic graph prediction. Although the absolute performance does not consistently outperform the state-of-the-art in every setting, and while the bridge between theoretical results and empirical gains could be further tightened, the discussion period significantly improved the quality of the paper. The AE finds that the main claims regarding the model's mechanism and efficiency are now convincingly supported by the provided evidence. Therefore, the AE recommends the paper for acceptance.

**Audience:**

Yes

**Audience Explanation:**

The paper addresses the problem of memory mechanisms in temporal graphs, a topic of clear interest to the TMLR audience, specifically those working on dynamic networks, state-space models, and memory-augmented architectures.

**Claims And Evidence:**

Yes

**Claims Explanation:**

After the discussion and revision, the submission provides claims that are accurately supported by evidence. The authors successfully narrowed the scope of their claims to align with the empirical results and provided additional experiments that better validate the approach.